# Consistent diel activity patterns of forest mammals among tropical regions

**Andrea F. Vallejo-Vargas**[1] ✉, **Douglas Sheil** [1,2,3], **Asunción Semper-Pascual**[1], **Lydia Beaudrot** [4], **Jorge A. Ahumada** [5], **Emmanuel Akampurira**[6,7], **Robert Bitariho** [7], **Santiago Espinosa** [8,9], **Vittoria Estienne** [10], **Patrick A. Jansen** [2,11], **Charles Kayijamahe**[12], **Emanuel H. Martin** [13], **Marcela Guimarães Moreira Lima**[14], **Badru Mugerwa** [15,16], **Francesco Rovero** [17,18], **Julia Salvador**[19], **Fernanda Santos**[20], **Wilson Roberto Spironello**[21], **Eustrate Uzabaho** [12] & **Richard Bischof** [1]

An animal's daily use of time (their "diel activity") reflects their adaptations, requirements, and interactions, yet we know little about the underlying processes governing diel activity within and among communities. Here we examine whether community-level activity patterns differ among biogeographic regions, and explore the roles of top-down versus bottom-up processes and thermoregulatory constraints. Using data from systematic camera-trap networks in 16 protected forests across the tropics, we examine the relationships of mammals' diel activity to body mass and trophic guild. Also, we assess the activity relationships within and among guilds. Apart from Neotropical insectivores, guilds exhibited consistent cross-regional activity in relation to body mass. Results indicate that thermoregulation constrains herbivore and insectivore activity (e.g., larger Afrotropical herbivores are ~7 times more likely to be nocturnal than smaller herbivores), while bottom-up processes constrain the activity of carnivores in relation to herbivores, and top-down processes constrain the activity of small omnivores and insectivores in relation to large carnivores' activity. Overall, diel activity of tropical mammal communities appears shaped by similar processes and constraints among regions reflecting body mass and trophic guilds.

Diel activity patterns—how animals distribute their activity throughout the 24 h day—vary among and within species[1]. Some species and individuals maintain activity over extended periods while others exhibit brief peaks of activity[1]. Animals may be predominantly active at night (nocturnal), day (diurnal), twilight (crepuscular), or may lack pronounced nocturnal or diurnal peaks (cathemeral). These activity patterns reflect when organisms seek food, socialize, and perform other necessary tasks while also responding to risks and physiologic constraints[2,3]. How these underlying processes and constraints shape activity patterns has been studied in various contexts, yet their identification at the community level, and their generality among regions has remained scarce due to a dearth of comparable data.

Mammals possess diverse specializations, including morphological, physiological, and behavioural adaptations that reflect and influence their diel behaviours[4]. These adaptations, including eye forms[5], sensorial systems, and endothermy (i.e., generation and regulation of body temperature) evolved in response to various needs and constraints (e.g., light, temperature, predation risk). Endothermy facilitates activity during cold periods[6], and may have benefitted early mammals by permitting nocturnal activity to reduce predation by diurnal dinosaurs[7]. Furthermore, interactions between physiological

---

characteristics, body size, and morphology may favour activity schedules that moderate exposure to thermal stress[8]. Large species may avoid overheating by limiting activity during warmer periods of the day[9,10]. By contrast, smaller species that can lose heat rapidly may favour activity in warmer periods of the day[11,12]. Moreover, activity patterns likely reflect a combination of processes and constraints. For example, small rodents may avoid diurnal predation through nocturnal behaviour, yet be active during daylight in response to food availability, temperature variation, or reduced competition or predation[2,13,14].

Species interactions—predation, competition—likely influence diel activity patterns within communities[15,16], yet, we lack a general understanding of how such interactions shape activity patterns. For instance, predators may favour periods where their prey are active, whereas prey species may avoid periods when their predators are active[17–19]. In other words, activity patterns could result from both top-down and bottom-up behavioural processes[2], analogous to the top-down and bottom-up consumptive processes that regulate food webs[20–22]. In a top-down process, one group of species (e.g., prey) adjusts their activity to avoid interacting with another group (e.g., predators or dominant competitors)[19,23]. For example, small carnivores may alter their activities to reduce their encounters with larger carnivores; similar avoidance behaviour is expected for prey (e.g., herbivores) to avoid their predators[18,23]. In a bottom-up process, on the other hand, predators may adjust their activity to facilitate encounters with their prey[24]. For instance, in four study areas in southwestern Europe, mesopredators match their activity to that of rodent prey[25]. Current evidence for bottom-up and top-down control of behaviour is restricted to scattered cases, regions, and communities[23–25]. For example, a top-down process was detected in African savannas where intermediate size-herbivores shifted their activity towards daytime when predation risk was high during the night[10]. The relative roles of top-down and bottom-up processes in shaping diel activity in mammal communities and the consistency of these processes among regions and biotas, therefore, remain uncertain.

Humid tropical forests provide an important context for exploring whether patterns in diel activity—thus potentially their main determinants—transcend biogeographical regions. In humid tropical forests the influence of seasonality is low, the environmental conditions across distinct regions are similar[8], and the maintenance of high species richness likely involves diverse interactions[26]. The trophic composition of mammal communities has been shown to be relatively consistent among regions[27]. If diel activity patterns are influenced by the same underlying processes as trophic guild composition, then we would expect consistency in diel activity patterns among regions.

Here, we study the diel activity patterns of ground-dwelling and scansorial (i.e., adapted to climb) mammals inhabiting protected tropical forests across the Neotropics, Afrotropics, and Indo-Malayan tropics. We examine patterns and test predictions associated with three alternative hypotheses (Fig. 1) for the main processes potentially

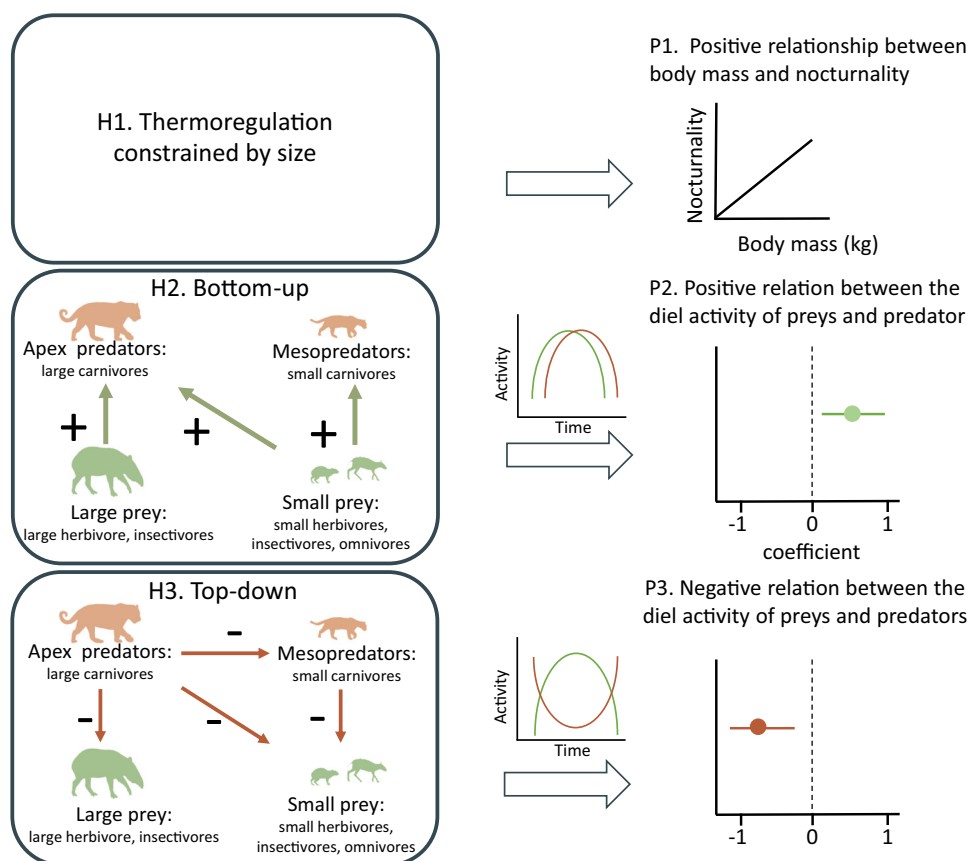

**Fig. 1 | Hypotheses (H1–H3) to determine processes that shape diel activity patterns in tropical forest mammal communities, with associated predictions (P1–3).** If the energetic cost of thermoregulation dominates (H1), we expect a positive relationship between body mass and nocturnality (1), regardless of trophic guild. If bottom-up regulation dominates (H2), predators will follow the diel activity of their prey (2). If top-down regulation dominates (H3), then we predict that small predators and potential prey species (herbivores and insectivores) will avoid top-predators (3). "+" represents a positive relationship between the activity of species groups (bottom-up process), and "−" represent a negative relationship between the activity of species groups (top-down process). Silhouettes from phylopic.org: jaguar, ocelot, and agouti by Gabriela Palomo-Munoz; tapir no license; browsing ruminant by Nobu Tamura (vectorized by T. Michael Keesey) http://creativecommons.org/licenses/by/3.0/.

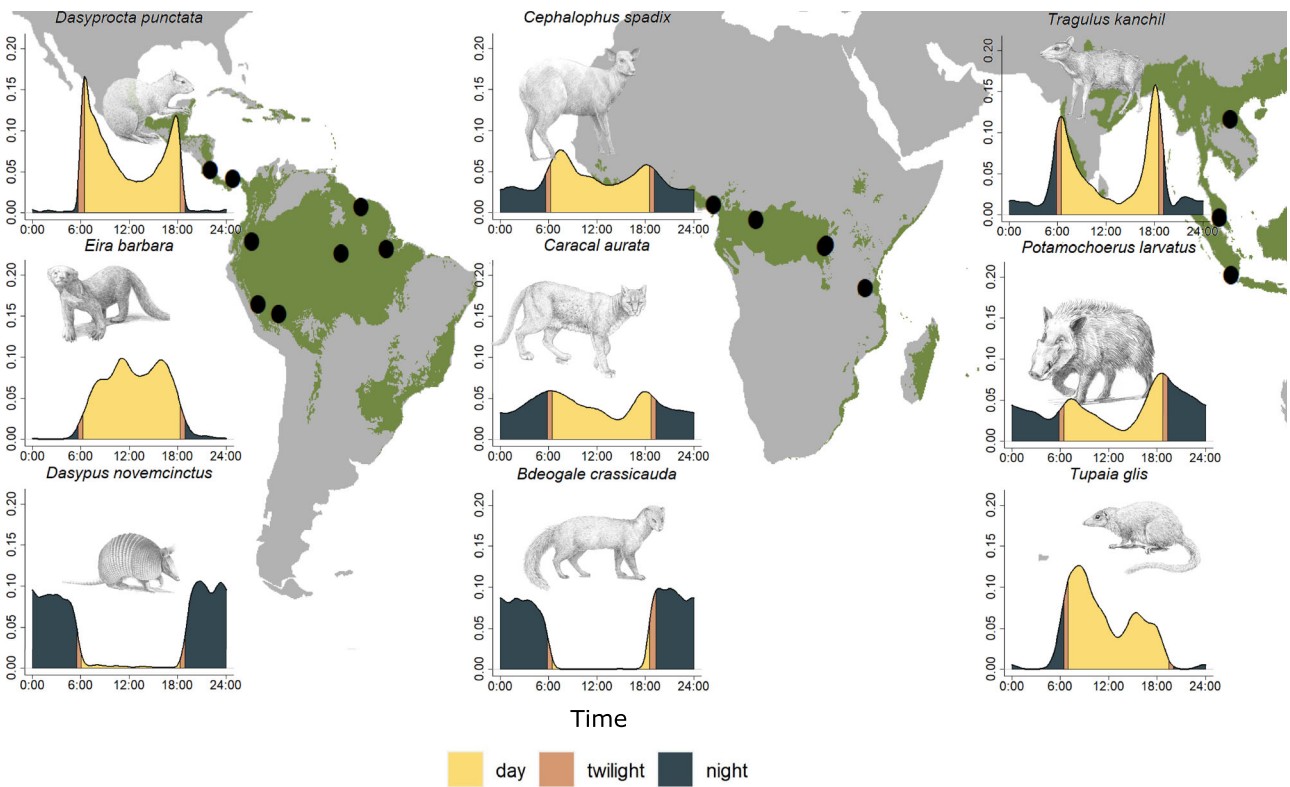

**Fig. 2 | Map of the study areas and activity density examples.** Mammal activity data were collected using the standardized TEAM camera-trapping protocol in 16 protected areas (black dots in background) situated in 14 countries and tropical forests (areas shaded green on the map in the background) in three biogeographic regions. Activity density plots in each column show examples of species in each region (from left to right: Neotropics, Afrotropics, and Indo-Malayan tropics). Illustrations by John Meaghan.

driving them. First, if the energetic cost of thermoregulation constrains diel activity (H1), then (1) larger mammals should be more active during the night when it is colder and smaller mammals more active during the day when it is warmer, irrespective of the dietary functional group. If bottom-up processes regulate diel activity (H2), then activity patterns of predators (e.g., carnivores) should match that of prey species (e.g., herbivores, insectivores). Finally, if top-down processes regulate the diel activity of animals in a community (H3), (3a) prey species such as herbivores should exhibit diel activity patterns contrasting those of predators of a similar size, and (3b) small carnivores should exhibit diel activity patterns that avoid large carnivores (Fig. 1). Here, we examine the diel activity pattern of distinct forest mammal communities using standard data collected from multiple sites across multiple regions. We show that diel activity appears remarkably consistent in relation to trophic guilds and body mass, which implicates multiple factors. First, herbivore activity and insectivores in two regions appears to be determined by thermoregulation. Second, smaller prey species (i.e., insectivores, and omnivores) and small carnivores reflect some top-down avoidance of top predators. Third, top-predators show bottom-up regulation of their activity in response to herbivores prey.

## Results

We used time-stamped images from standardized large-scale camera-trap surveys implemented by the Tropical Ecology Assessment and Monitoring (TEAM) Network in 16 protected areas (Fig. 2 and Table S1)[28] to examine and test our hypotheses. First, to identify if there were consistent patterns across regions, we used multinomial analysis with random intercepts (protected area) for each biogeographical region to investigate how diurnal, nocturnal, and crepuscular activity was related to the trophic guild and body size. The best model

based on the lowest Akaike information criterion (AIC) contained an interaction between body mass and guild and best explained the activity of mammals in all regions. We extracted the probability of being active during the day, night, and twilight, and the correspondent upper (UCI) and lower (LCI) 95% confidence intervals for the given range of body mass and trophic guild derived from the best multinomial model. Second, to test how top-down and bottom-up processes shape diel activity, we divided species into small and large categories for each trophic guild and tested whether the hourly activity of prey (e.g., large herbivores) or subordinate species (e.g., small carnivores) was correlated with the activity of predators (e.g., large carnivores). We tested the top-down and bottom-up hypotheses for all protected areas where top predators had been detected ($N = 11$, Table S1), and utilized generalized linear mixed models (GLMM) with the protected area as a random intercept. Positive coefficients were interpreted as an overlap of activity, while negative coefficients were interpreted as a temporal avoidance between the activity of the groups compared. We further assessed how top-down, and bottom-up processes shaped the diel activity of tropical mammals by plotting the density distribution of all species groups (prey/subordinate species vs. predators) and estimating the coefficients of overlap ("Dhat", see "Methods") for each protected area. This coefficient ranges from 0 to 1 with higher and lower values interpreted as bottom-up and top-down influences, respectively.

### Consistent patterns of diel activity

Diel activity, as analyzed with multinomial models, was generally well explained by the interaction between body mass and trophic guild in all three regions (Fig. 3 and Tables S2, S3), despite substantial variation in diel activity patterns among species (Figs. 1 and S4). The probability of nocturnal activity by herbivores increased with increasing body

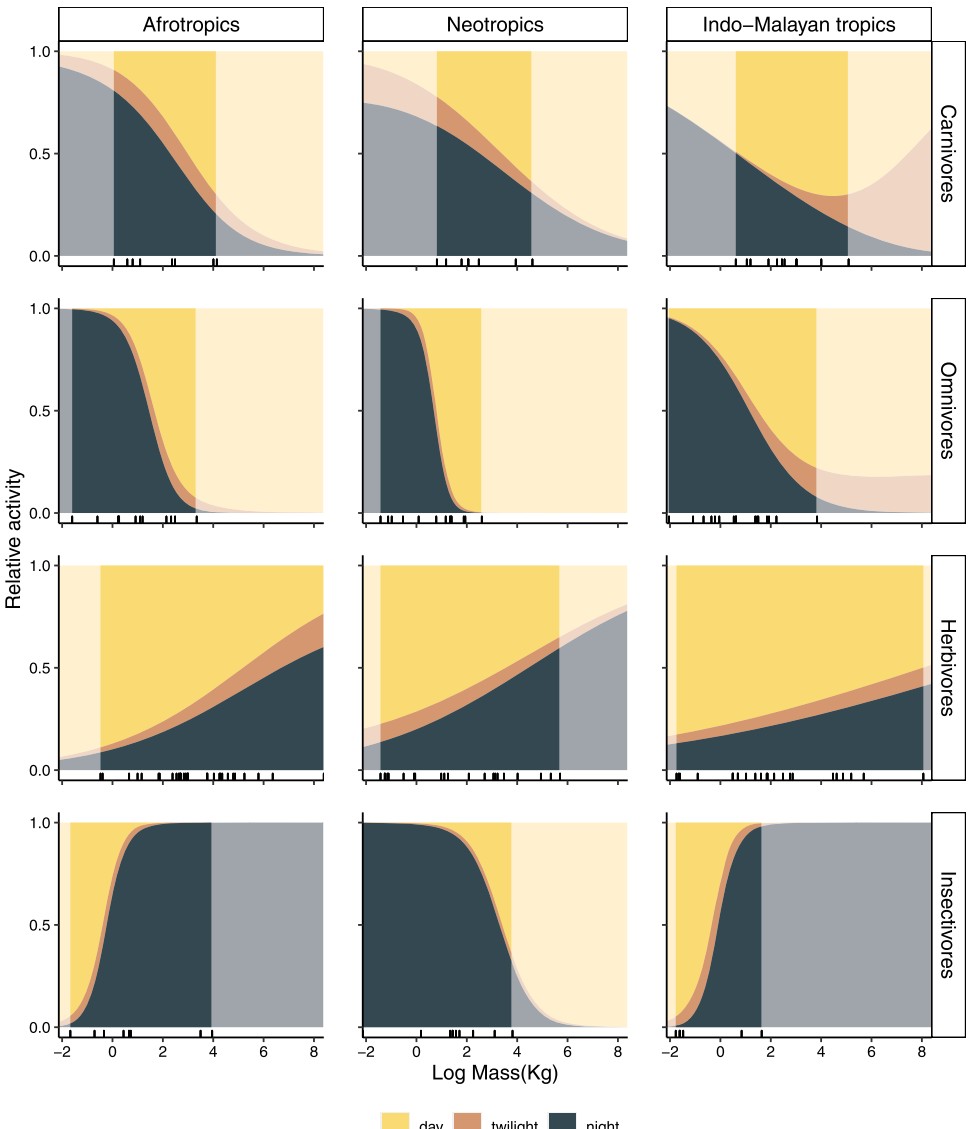

**Fig. 3 | Diel activity in relation to body size and trophic guilds of tropical ground-dwelling and scansorial mammals in three regions.** Estimates correspond to the probability of activity during the day, night, and twilight extracted from the multinomial logit models fitted to TEAM camera-trap data ($n = 126,382$). Tick marks above the $x$-axis indicate the body mass of species included in the analysis. Lighter colours indicate model predictions for body masses that are below or above the range for species included in the analysis in each region. "$n$" represents the number of independent events. $n_{carnivores\_Neotropics} = 2182$, $n_{carnivores\_Afrotropics} = 1474$, $n_{carnivores\_Indo-Malayan\_tropics} = 152$, $n_{omnivores\_Neotropics} = 4656$, $n_{omnivores\_Afrotropics} = 4656$, $n_{omnivores\_Indo-Malayan\_tropics} = 435$, $n_{herbivores\_Neotropics} = 45,839$, $n_{herbivores\_Afrotropics} = 47,458$, $n_{herbivores\_Indo-Malayan\_tropics} = 7803$, $n_{insectivores\_Neotropics} = 4399$, $n_{insectivores\_Afrotropics} = 3886$, $n_{insectivores\_Indo-Malayan\_tropics} = 212$.

mass in all regions (Fig. 3). For example, the largest herbivore in the Neotropics was 4.6 times more likely to be nocturnal than the smallest herbivore (e.g., large: $p_{night} = 0.60$, CI: 0.48–0.71, body mass = 210 kg; small: $p_{night} = 0.13$, CI: 0.08–0.21, body mass = 0.24 kg, Fig. 3). The opposite relationship occurred for carnivores and omnivores in all regions. For example, a 61 kg carnivore in the Afrotropics was 3.9 times less likely of being active at night ($p_{night} = 0.21$, CI: 0.14–0.28) than a 1 kg carnivore ($p_{night} = 0.81$, CI: 0.74–0.87).

Insectivores in the Neotropics were an exception from the general pattern (Fig. 3, Fig. S1, and Table S2). Whereas Afrotropical and Indo-Malayan insectivores exhibited a positive relationship between body mass and the probability of nocturnal activity (e.g., in the Indo-Malayan region nocturnal probability increased from 0.01 to 0.98), in the Neotropics nocturnality decreased with increasing body mass, from a probability of 0.99 (CI: 0.99–0.99, body mass = 0.12 kg) to 0.32 (CI: 0.22–0.44, body mass = 43.30 kg, Fig. 3).

## Thermoregulation constrains the activity of herbivores and insectivores

The positive relation between nocturnality and body mass for herbivores and insectivores (Afrotropics and Indo-Malayan tropics) was congruent with the prediction for H1. Nevertheless, carnivores, omnivores, and insectivores in the Neotropics showed the opposite relationship.

## Top-down and bottom-up processes shape the diel activity of tropical mammals

Our GLMM analyses of the relationship between the activity of different trophic groups and different sizes (large and small) suggests that a combination of bottom-up (H2) and top-down (H3) processes shaped the diel activity of mammalian groups among regions. Consistent with H2 (bottom-up), we found evidence of a positive relationship between the activity of large herbivores and large carnivores across the three

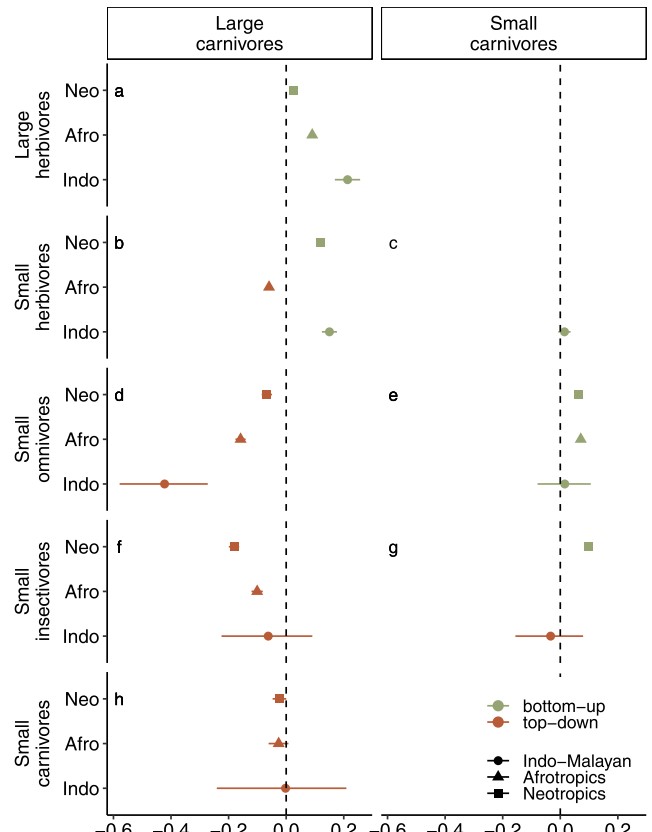

**Fig. 4 | Bottom-up and top-down processes as determinants of the diel activity of tropical mammals.** Centre of bars represent the mean coefficient estimates and bars show the 95% confidence intervals of the (GLMM) fitted to assess the relationship between the activity of species groups. The first column includes the relationship between the activity of large carnivores ($n = 747$) and prey (**a** large herbivores $n = 191,294$, **b** small herbivores $n = 58392$, **d** small omnivores $n = 8098$, and **f** small insectivores $n = 7120$) and **h** the relationship between the activity of large carnivores and small carnivores ($n = 2280$). The second column includes the relationship between small carnivores and potential prey (**c** small herbivores, **e** small omnivores, and **g** small insectivores). Note that $n$ represents the total number of independent events for each species group and size. Green symbols illustrate a positive effect (bottom-up) and brown symbols illustrate a negative (top-down) relationship. Effects were considered significant when the 95% CI did not overlap zero (dashed horizontal lines). Neotropical sites "Neo" are denoted with squares, Afrotropical sites "Afro" with triangles, and Indo Malayan "Indo" with circles.

regions studied (e.g., Neotropics: $\beta = 0.03$, CI: 0.02–0.04; Indo-Malayan: $\beta = 0.21$, CI: 0.17–0.26, Fig. 4a). Similarly, we detected a positive relationship between the activity of small herbivores and the activity of large carnivores in the Neotropics and Indo-Malayan tropics (e.g., Neotropics: $\beta = 0.12$, CI: 0.13–0.13, Fig. 4b). The activity of small carnivores in the Afrotropics and Neotropics exhibited a significant positive relationship with the activity of small omnivores (e.g., Afrotropics: $\beta = 0.07$, CI: 0.07–0.07, Fig. 4e) and small insectivores in the Neotropics ($\beta = 0.10$, CI: 0.09–0.11, Fig. 4g). Inconsistent with the bottom-up hypothesis, the activity of large carnivores vs. small herbivores showed a negative relationship (Fig. 4b) in the Afrotropics.

Consistent with top-down processes (H3), we detected a negative relationship between the activity of large carnivores vs. small omnivores across all regions (Fig. 4d) and for the activity of large carnivores vs. small insectivores in two regions as indicated by the GLMMs (Neotropics, $\beta = -0.18$, CI: −0.20 to −0.16; Afrotropics: $\beta = -0.10$, CI: −0.12 to −0.09, Fig. 4f). Additionally, albeit no-significant support for H3 was suggested by the GLMM, the activity of small and large carnivores tended to be negatively correlated (Fig. 4h).

Overlap estimates varied depending on the species groups compared as well as the protected area. The lowest variability among protected areas was found for the overlap estimates between the activity of large carnivores and large herbivores (10 out of 11 protected areas was higher than 0.78, CI:0.67–0.82, Fig. S5). These results provide support for the bottom-up hypothesis (H2). In contrast, the overlap estimates for the rest of the species group comparisons were less consistent (Figs. S6–S12). For example, overlap estimates between the activity of small omnivores and large carnivores ranged from Dhat1 = 0.39 (CI: 0.29–0.5) to Dhat4 = 0.85 (CI: 0.76–0.92, Fig. S8).

We did not detect significant relationships between the activity of large insectivores and large predators, and the data were too sparse to include models comparing large omnivores with other groups.

## Discussion

Our study revealed similar relationships of trophic guild and body mass with diel activity patterns of tropical forest mammals in distant biogeographic regions despite the variation in species-specific activity patterns (Fig. S3). These results suggest convergent ecological and/or evolutionary responses in diel activity among tropical regions. Such convergence, despite the considerable taxonomic differences in regional biotas, likely reflects the results of adaptations to similar environments. Among carnivores and omnivores, larger species were less likely to be nocturnal than smaller ones. In contrast, larger herbivores, tended to be more nocturnal. Insectivores were an exception because they showed a negative relationship between body size and nocturnality in the Neotropics but a positive relationship in the Afrotropics and Indo-Malayan regions.

Despite the overall consistency in diel activity patterns across the pantropics, our analysis did not point towards a single dominant driver for the observed patterns. Instead, it appears that multiple factors may have acted simultaneously. Thermal constraints (H1), bottom-up (H2), and top-down (H3) processes all seemed to contribute to the configuration of activity within tropical forest mammal communities (Figs. 3, 4). Increasing nocturnality with body mass for herbivores and insectivores (Afrotropics and Indo-Malayan tropics) is consistent with the hypothesis on thermoregulatory constraints (H1). Furthermore, trophic interactions, known to influence species richness and biodiversity[26,29], appear in our study to be important influences on diel activity patterns through both top-down and bottom-up processes. Although multiple factors (e.g., predation risk, prey abundance) appear to have influenced interactions, there was nonetheless some uniformity observed among regions. Carnivores tended to match the diel activity of potential prey species, supporting the bottom-up hypothesis (H2). On the other hand, in some regions the activity of small insectivores, small omnivores, and small carnivores was best explained by the top-down hypothesis because these groups seemed to avoid periods when larger carnivores were active (H3).

Consistent with the thermoregulatory constraint hypothesis (H1), we found that larger-bodied herbivores and insectivores were more likely to be nocturnal than smaller-bodied ones. While diel temperature is more stable in tropical rainforests than in many other ecosystems, it does vary[30]. Most tropical mammals are adapted to survive in a narrow thermal tolerance range[31,32], thus both high and low temperatures can increase energy expenditure[33]. Small-bodied species can reduce energy loss by being active during warmer periods of the day[11], while large-bodied animals (e.g., tapirs[34], aardvark[35]) can reduce thermal stress by focusing their activity during cooler periods of the day[9,34,36]. For example, in the Neotropics the probability of being active during the night was two times higher for a 290 kg herbivore (e.g., *Tapirus bairdii*) than for one weighing 1 kg (e.g., *Myopracta acouchi*).

If thermoregulatory constraints were the sole or primary driver of diel activity, we would anticipate the relationship between mass and activity to manifest across all trophic guilds and regions. This was not the case. Carnivores and omnivores did not exhibit a positive

relationship between size and diurnality. This may in part be explained by the lack of large species in those groups or less severe risk of thermal stress. Alternatively, our study suggests that there is a greater role of species interactions (bottom-up and top-down processes) influencing diel activity patterns for carnivores and omnivores in humid tropical forests. Another group exhibiting behaviours inconsistent with the thermoregulatory constraint hypothesis was the Neotropical insectivores. The higher diurnal activity of larger versus smaller Neotropical insectivore species was dominated by just three species (*Myrmecophaga tridactyla, Tamandua tetradactyla, and Tamandua mexicana*)—all of which reflect the distinct South American native lineages that persisted after the great interchange[37]. The different behaviour in this group may be due to chance, the low number of species, or characteristics neglected by our guild categories. For example, among large insectivores, Neotropical anteaters live above ground unlike the fossorial aardvarks of the paleotropics. Another possibility beyond the scope of our current study is that there may be differences in the presence and temporal availability of insect prey.

The positive correlation in the diel activity of large carnivores and large herbivores was relatively consistent among regions (Fig. 4) and overlapped more than expected by chance among protected areas (Fig. S6). Similarly, small carnivores seemed to match their activity to that of small potential prey (e.g., small omnivores and small insectivores, Fig. 4). We infer that these carnivores sought to increase encounters with prey. Previous studies have reported a similar match between predator and prey activity[25,38–40]. For example, the activity of the Borneo Sunda clouded leopard (*Neofelis diardi*), a top-predator, overlaps with its preferred prey species, the sambar deer (*Rusa unicolor*) and small herbivore greater mouse deer (*Tragulus napu*)[41]. We also found evidence to the contrary: the activity of small herbivores in the Afrotropics indicated temporal avoidance of large carnivores (Fig. 4b), potentially due to the abundance or richness of prey or predator species in the Afrotropics. For example, when predator abundance increases, prey have been observed to adjust their activity to reduce interactions with predators[23]. We speculate that the temporal avoidance we reported in the Afrotropics may reflect lower prey availability or higher predator abundance that resulted in higher predation risk and a resulting shift in the activity of herbivore prey. We do not have reliable estimates on abundance to evaluate these nuances directly.

Our analysis revealed apparent temporal avoidance of the activity of large carnivores by small omnivores in the Indo-Malayan tropics and Afrotropics and by small insectivores in the Neotropics and Afrotropics. Avoidance of large carnivores could decrease antagonistic interactions (e.g., predation, interguild killing) with large predators[19,42], which exert top-down behavioural control. We detected a signal of temporal avoidance from the negative relationship between the activity of small and large carnivores in two regions (Neotropics and Indo-Malayan tropics) consistent with previous studies demonstrating temporal avoidance among species pairs. For instance, an earlier study[43] in some of our Neotropical study areas, revealed that ocelots (*Leopardus pardalis*) exhibited a low overlap with the activity of the larger jaguar (*Panthera onca*) and puma (*Puma concolor*). The present study suggests that, overall, the activity of smaller carnivores in protected tropical forests is to a large extent motivated by bottom-up processes (H2)—i.e., facilitate encounters with potential prey such as small omnivores and insectivores—rather than top-down processes (H3)—i.e., avoidance of intraguild interactions with larger carnivores. Nonetheless, there is likely substantial variation among species in the relative importance of top-down and bottom-up processes, with both potentially playing a role. For example, ocelot activity overlaps with various omnivorous prey species, such as opossums, raccoons[44], insectivores as armadillos[45], while it also avoids jaguars[43].

Despite some consistency between the GLMM and the overlap analysis, there was also variation between them. For example, comparing the activity of large carnivores and herbivores, most protected areas exhibited high overlap coefficients consistent with the bottom-up hypothesis (H2), yet one protected area differed (e.g., Manaus, Fig. S6). In other cases, the overlap coefficients among protected areas varied greatly and limited us from inferring general mammalian diel activity patterns. Thus, the use of GLMM allowed a more formal assessment of bottom-up and top-down processes at the regional level while accounting for variation among protected areas.

Although all study areas were relatively well-protected, none were completely free of human impacts[28] raising the question of how this may have influenced our observations. Human presence and activities can have pronounced impacts on wildlife activity; for example, species may become more nocturnal to avoid hunters[46]. This has been observed in Yasuní, one of our study areas, where ungulates became more nocturnal as hunting increased[47]. Our study cannot clarify the role of hunters in determining the specific details of our results and we are wary of such attempts. Simple approaches using human activity may be misleading as evasive responses among mammals are not universal and can change over time (for example, the gorillas in Bwindi have been habituated to humans). At some of the study areas, certain large predators that were previously present are now scarce or absent (e.g., leopards in Bwindi[48])[49,50], raising questions concerning how the prey community (e.g., omnivores and insectivores) may respond.

Despite distinct origins, biogeographic histories, and taxonomic compositions, community level diel activity patterns for tropical forest mammals exhibited consistent patterns in relation to trophic guild and body size across three tropical biogeographic regions. Convergent responses—ecological and/or evolutionary—to similar conditions among regions appear manifested in similar diel activity strategies within these diverse communities. Furthermore, our analysis pinpoints different determinants depending on trophic guild. Herbivore and insectivore activity appears to be shaped by thermoregulatory constraints while predator-prey interactions appear to be influenced by the temporal behaviour of their members. Thus, bottom-up processes dominate the activity of carnivores, and top-down processes dominate the activity of prey (mainly omnivores and insectivores).

## Methods

### Study areas and camera trapping

We used camera-trap data from the Tropical Ecology Assessment and Monitoring (TEAM) Network[49]. TEAM data comprise data from three tropical biogeographic regions (Neotropics, Afrotropics, and Indo-Malayan tropics) and 16 protected areas (TEAM Network, 2011) (Fig. 1). Camera-traps were deployed following a standardized protocol in all protected areas during the dry seasons between 2008 and 2017. At each protected area, the monitoring ran from two to ten years with the deployment of 60 to 90 cameras annually. Camera-traps were placed at a density of 0.5–1 camera/km² (1 camera every km² or 1 camera every 2 km²) and remained active for ~30 consecutive days[28,49]. We excluded data from camera-trap sites with inconsistent date-time stamps, yielding a total of 60–89 cameras per protected area (Fig. 1 and Table S1).

### Data

A total of 2,312,635 camera-trap photos included mammals. We further filtered the dataset to include only species with a body mass greater than 75 g (smaller species have high uncertainty of identification and are difficult to detect) and strictly terrestrial or scansorial species (i.e., we excluded all arboreal and aquatic species)[27,51]. A total of 166 species, 38 families, and 15 orders of ground-dwelling and scansorial species were included in our study (Table S1). Since camera-traps often take multiple consecutive pictures of the same visit or individual, we avoided pseudo-replication of individuals by establishing independent events (time interval between pictures > 1 h per camera for a given species). This resulted in a total of 126,382 independent events. To analyze diel activity, we used the time-stamp recorded in each

independent event[52]. To test whether activity was consistent among tropical regions and to test H1, we summarized the number of events for each of the following three categories (1) day, (2) twilight, or (3) night. Each event was classified by protected area, location, time, and date to specify the sunrise, sunset, nautical dawn, and dusk using the R library 'maptools' version 1.1–4[53] and the functions 'crepuscule' and 'sunriset'. Twilight was defined as the interval between dawn and sunrise and between sunset and "nautical dusk"[54]. Day was defined as the interval between sunrise and sunset. Night was the interval between nautical dusk and nautical dawn. To test H2 and H3, and to plot species-specific activity profiles, every independent event was anchored to sunrise and sunset to correct for differences in the delimitation of day, night, and twilights between protected areas and across seasons[55] using the 'activity' package[56,57].

We extracted (1) diet, (2) body mass (g), and forest strata from the PHYLACINE database[58] and updated reviewed data on forest strata of mammals in the protected areas studied[51] (Fig. S2). We excluded the arboreal species and only included ground-dwelling and scansorial species in our study. Then, we classified each mammal species into one of four trophic guilds: carnivore, herbivore, insectivore, or omnivore. Categories were based on diet reported in the PHYLACINE database and we classified as carnivore species feeding on ≥80% vertebrates, herbivore species feeding on ≥80% plant materials, insectivore feeding on ≥80% insects, the remaining species were categorized as omnivores (e.g., feeding on vertebrates and fruits)[58,59].

## Analysis

To test how trophic guild (discrete variable: carnivores, herbivorous, insectivores, and omnivores) and body mass (continuous variable: log-transformed) were associated with the number of independent events of each diel activity (day, night, twilight) of tropical ground-dwelling and scansorial mammals we fitted a multinomial logit model[60] using the package 'mclogit' version 0.9.4.2[61]. Multinomial modelling allowed us to assess three response classes (day, night, and twilight), as opposed to two responses classes in logistic regression models. We fit a set of candidate models for each tropical region (Neotropics, Afrotropics, Indo-Malayan tropics) using maximum likelihood (ML) and with a convergence tolerance (Ɛ) of 1e−6 (Table S1). To account for potential non-independence in activity patterns of species detected in a given protected area, we included protected areas as a random intercept effect within all models. We selected the best model for each tropical region using Akaike information criterion (AIC)[62]. We ranked models using ΔAIC and considered models with a ΔAIC < 2 to be equally supported. Once we selected the best models, we ran the models with a restricted maximum likelihood (REML) to arrive at final estimates for each tropical region. We predicted relative activity with the package 'mpred' version 0.2.4.1[61]. This allowed us to extract the predicted probability of activity falling into each diel category for the range of body masses, for each trophic guild, and region.

To test if the diel activity of tropical mammals showed indication of arising from top-down or bottom-up processes, we classified trophic guilds by size to test how the hourly activity (number of independent events), anchored to sunrise and sunset, of large and small groups (cut-up of 20 kg[63]) respond to the activity of large and small predators. We excluded species with very low risk of predation, the African buffalo *Syncerus craffer*, and elephant species[64] (body mass >580 kilograms). We used a log link and a Poisson distribution in package "lme4" version 1.1–29 for each region to assess the relationship between the activity of a) large and small herbivores, insectivores, omnivores, carnivores (response variable) and b) large and small carnivores (predictor variable). Significant negative and positive model coefficients were interpreted as evidence for top-down and bottom-up effects, respectively. We did not include the comparison between large omnivores and large carnivores in our models because there were not sufficient detections to test this combination. We also excluded

models that did not converge (small carnivores vs. small herbivores in the Neotropics and Afrotropics, and small carnivores vs. small insectivores in the Afrotropics). We employed the data of 11 protected areas where large carnivores were present (Table S1) and set protected area in the models as a random intercept.

In addition, we plotted the kernel density distribution of the activity of each trophic guild and size and (e.g., prey-predator) extracted the overlap estimates in each protected area to exemplify our results from the GLMM models assessing the bottom-up and top-down processes on diel activity. To compare the activity of prey species (e.g., herbivores) and predators (i.e., carnivores) with different sizes, we extracted the coefficient of overlap (Δ "Dhat") between the two kernel density distributions with the package 'overlap' version 0.3.4[65]. If the sample size was ≥75 independent events, we extracted the coefficient of overlap type "Dhat1", if the sample size was higher than 75 we extracted the "Dhat4"[66]. In addition, we tested the probability that the fitted distributions of the activity among pairwise groups (e.g., large herbivores vs. large carnivores) came from the same distribution by employing 500 bootstrap iterations, and obtained 95% confidence intervals (CI) and the 'probability observed index arose by chance' (*P* pNull) using the package 'activity' version 1.3.2[57]. Low values of this coefficient indicate avoidance between groups of species and *P* is the probability that the overlap between groups arose by chance (Supplementary Material PDF). It is worth mentioning that, we did not run a regional model to extract the coefficient of overlap among groups of species because pooling data from different study areas may overestimate the coefficient of overlap and lead to biased inferences[66].

To plot the activity patterns of species from Fig. 2 and Fig. S3, we gathered the data of all protected areas in each tropical region and characterized species-specific activity patterns when the number of independent events was 25 or more[66] (Fig. S3). Then we plotted species activity with the package 'overlap', which employs kernel density estimation that circumvents the conflation of data required for histograms[66]. The map for Fig. 2 was prepared in ArcGIS 10.8.1, and the composed Fig. 2 was prepared in Inkscape 1.1.1. All statistical analyses and plots were made in R-4.2.1[67].

## Reporting summary

Further information on research design is available in the Nature Portfolio Reporting Summary linked to this article.

## Data availability

The data generated in this study have been deposited in the DataverseNO database is available online at https://doi.org/10.18710/BIGEO7. The raw camera-trap data employed in this study can be found in Wildlife Insights (www.wildlifeinsights.org). Species characteristics extracted from PHYLACINE 1.2 are available online at https://doi.org/10.5061/dryad.bp26v20. Species list with reviewed forest strata data are available at https://doi.org/10.5061/dryad.f1vhhmgv0.

## Code availability

The code to analyze and reproduce this study has been deposited in the DataverseNO and is available online at https://doi.org/10.18710/BIGEO7.

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

## Acknowledgements

We thank the funding by Research Council of Norway (project NFR301075 to D.S., and R.B.), and National Science Foundation grant (DEB-2213568 to L.B.). This work was made possible by the Tropical Ecology Assessment and Monitoring (TEAM) Network, a collaboration between Conservation International, the Smithsonian Tropical Research Institute, and the Wildlife Conservation Society. We acknowledge the effort of all TEAM site managers and collaborators who helped collecting data as well as Wildlife Insight for the data processing and availability and David Kenfack. We acknowledge the suggestions of Pierre Dupont for the analysis. Finally, we thank John Megahan for species illustrations in Fig. 1.

## Author contributions

D.S. and R.B. proposed the study and accessed funding. A.F.V.-V., R.B., and D.S. developed the approach and hypotheses presented here. A.F.V.-V. developed and performed the analyses. R.B. verified the analysis. A.F.V.-V. wrote the manuscript with support from R.B., D.S., A.S.-P., and L.B. The authors D.S. J.A., E.A., R. Bitariho, S.E., V.E., P.A.J., C.K., E.H.M., M.G.M.L., B.M., F.R., J.S., F.S., W.R.S., and E.U. were responsible for camera trap data collection in the TEAM study areas. A.F.V.-V., R.B., and D.S. finalized the manuscript with input and approval from all authors.

## Competing interests

The authors declare no competing interests.

## Additional information

[1]Faculty of Environmental Sciences and Natural Resource Management, Norwegian University of Life Sciences, 1432 Ås, Norway. [2]Department of Environmental Sciences, Wageningen University and Research, Wageningen, The Netherlands. [3]Center for International Forestry Research (CIFOR), Kota Bogor, Jawa Barat 16115, Indonesia. [4]Department of BioSciences, Program in Ecology & Evolutionary Biology, Rice University, Houston, USA. [5]Moore Center for Science, Conservation International, Arlington, VA, USA. [6]Department of Conflict and Development Studies, Ghent University, Sint-Pietersnieuwstraat 41, 9000

Ghent, Belgium. [7]Institute of Tropical Forest Conservation, Mbarara University of Science and Technology, P.O Box 44 Kabale, Uganda. [8]Facultad de Ciencias, Universidad Autónoma de San Luis Potosí, San Luis Potosí, México. [9]Escuela de Ciencias Biológicas, Pontificia Universidad Católica del Ecuador, Quito, Ecuador. [10]Wildlife Conservation Society, Congo Program, 151 Avenue General de Gaulle, Brazzaville, Republic of Congo. [11]Smithsonian Tropical Research Institute, Panamá, República de Panamá. [12]International Gorilla Conservation Programme, Kigali, Rwanda. [13]College of African Wildlife Management, Mweka, Department of Wildlife Management, P.O. Box 3031 Moshi, Tanzania. [14]Laboratório de Biogeografia da Conservação e Macroecologia, Instituto de Ciências Biológicas, Universidade Federal do Pará, Pará, Brazil. [15]Leibniz Institute for Zoo and Wildlife Research, Alfred-Kowalke-Straße 17, 10315 Berlin, Germany. [16]Department of Ecology, Technische Universität Berlin, Straße des 17. Juni 135, 10623 Berlin, Germany. [17]Department of Biology, University of Florence, Florence, Italy. [18]MUSE-Museo delle Scienze, Trento, Italy. [19]Wildlife Conservation Society Ecuador, Mariana de Jesus E7-248 y Pradera, Quito, Ecuador. [20]Programa de Capacitação Institucional, Coordenação de Ciências da Terra e Ecologia, Museu Paraense Emílio Goeldi, Belém Pará, Brazil. [21]Grupo de Pesquisa de Mamíferos Amazônicos, Coordenação de Biodiversidade, Instituto Nacional de Pesquisas da Amazônia, Manaus, Amazonas, Brazil. ✉e-mail: andrea.f.vallejo.vargas@nmbu.no

