## [Peer Review File · Nature Communications]

Reviewer comments, first round –

Reviewer #1 (Remarks to the Author):

The authors present an extensive and interesting review of activity patterns in forest tropical mammal communities. This paper capitalises on the valuable information of community patterns obtain from camera studies and takes advantage of the extensive TEAMS camera trapping network.

I found the paper interesting and think it would appeal to a general audience. I have some minor comments below:

1. I found the development of the hypotheses a bit confusion in parts - the use of terms top-down/bottom up are used in combination with intraguild/interguild interactions. I think using the former alone might make the arguments easier to understand. Some of these relationships may be related to dominance as well as predation. I would ask would we expect to see dominance influencing large-small herbivore interactions as well?
2. I would think we might expect some of these hypotheses more likely than others. The pred-prey intereactions is complicated as both predators and prey may be co-evolving strategies whereas, large predators may not be trying to map activities based on small predators so there is the clear opportunity for a joint strategy and more likely clearer results. Also this could then lead to clearerr strategies of small prey avoidance.. but then this is difficult to separate from temperature regulation of small species. I wonder if the authors could suggest if one hypothesis is dominant and likely to then lead to further activity patterns in these communities.
3. Labelling could be better in Fig 2.. can you put the H1 H2 etc on the plot? Avoid too many terms? Would you expect any interaction between small and large herbivores?
4. Fig 3. It might be nice to see individual activity charts for some of the species marked on the plots here. Perhaps in a separate panel to show how the predicted and observed ratios of activity compare.

Reviewer #2 (Remarks to the Author):

Reviewer comments to the manuscript "Consistent daily activity patterns across tropical forest mammal communities" by Vallejo-Vargas et al.

The authors report the quantification of diel organization of different tropical forest species guilds using data from standardized camera trapping protocols in 16 protected areas across three continents. The outcome is a remarkable consistency in the size-dependent temporal organization of daily activity in different trophic guilds. The authors pose three hypotheses on the variance in diel organization within and between these guilds and argue that trophic interactions provide the best explanation to support these. The manuscript is clear and well written; a pleasure to read. I have only minor comments.

Although the observation of homology in daily organization of activity in guilds across tropical rainforests is highly interesting and a strong contribution in itself, the explanation why this actually happens is more speculative. The authors argue this depends on generalized inter- and intra-guild trophic interactions. This reasoning potentially benefits from a quantification of variance in diel organization within (size classes of) guilds. Furthermore, the study strongly hinges on the assumption that larger species are always preying on smaller species, which is of course logical, but predator-prey interaction may not always depend on predator-prey size differences. Anyway, possible exceptions(?) in top-down interactions do not blur the size-driven diel activity organization. If variation in size-dependent predator-prey interaction can be quantified, this would be good. Anyway, putting these hypotheses forward is useful in itself.

Insectivores, line 250-252 – I assume the variation in temporal organization of insect prey is huge anyway, and cascades to the insectivore level?

Line 313 – what is meant with independent events here?

Reviewer #3 (Remarks to the Author):

Review for:

Consistent daily activity patterns across tropical forest mammal communities

Note: This review is written in markdown format. I've also attached a pdf if that is easier to read.

In this paper the authors assessed diel activity patterns of mammals in humid tropical forests across three biogeographical regions. I was very excited to get to read this manuscript, as it aligns with my own research interests.

Overall, this is a high quality piece of research that the authors should be proud of. I also commend the authors for using a multinomial model to quantify diel activity patterns, as it gave them the ability to more directly assess their hypotheses in relation to body size.

I only have what I would consider minor suggestions to this article, which hopefully will improve some parts that were a little confusing to me and improve the cohesion of this piece of research. If you have any specific questions about my comments, feel free to reach out to me directly (Mason Fidino, mfidino@lpzoo.org).

Following these two 'top-level' comments, I have smaller comments about various sections of the manuscript. I hope the authors find these suggestions helpful.

1. On the use of the phrase daily activity patterns versus diel activity pattern. Since the authors are using both terms, and they are effectively synonymous, I suggest that they just try to stay consistent throughout the text. My personal preference is for diel activity, as I feel it is a bit more specific, but the authors are free to choose whatever phrasing they like.

2. I am a big fan of using a multinomial model for diel activity patterns, as it allows you to incorporate continuous covariates to look at changes in diel activity patterns. However, one unique aspect of applying such a model to these data is that the temporal 'bins' for each time period are different amounts of time. What is great though, is that we know how much time is available for each diel period over 24 hours. If you wanted to convert your model into more of a temporal resource selection function, so you are quantifying use relative to availability, you could add a log offset term to the model for each diel period that is the amount of hours associated to each diel category. You should end up with essentially the same results (i.e., the slope terms should not change, only the intercepts). Right now the intercepts are a mix of use & availability, which is not ideal. See Gallo et al. (2022) for an example of a multinomial model with a log offset term.

...

Gallo, T., Fidino, M., Gerber, B., Ahlers, A. A., Angstmann, J. L., Amaya, M., ... & Magle, S. (2021). Mammals adjust diel activity across gradients of urbanization. bioRxiv.

...

Link to preprint:

<https://www.biorxiv.org/content/10.1101/2021.09.24.461702v1.abstract>

Summary

Top-level thoughts

1. I like that the abstract provides some direction of certain relationships the authors found (e.g., larger herbivores tended to be more nocturnal). If space allows, it would be great to add in some small info about effect size and the like here (e.g., larger herbivores were X times more likely to be nocturnal than smaller herbivores).

Line by line comments

Line 47: Very minor thing, but it reads a little weird to use 'specific' twice in the same sentence like this. Is the second 'specific' really needed?

Line 48: It takes a second to connect 'these patterns' to what I assume you mean 'daily activity patterns'? May help to just exchange 'these patterns' for 'diel activity patterns.'

Introduction

Line by line comments

Line 71: There are a lot of behaviours brought up in this paragraph, and so I'm not certain what 'such behaviours' refers to for mammals here. Also, adding mammals at the very end of this paragraph comes a bit out of left field. I think it may read better if you had a different crystallizing point at the end of this paragraph and then start the following paragraph with something like 'Mammals illustrate a broad range of diel activity patterns and occupy all temporal niches (day, night, twilight). However, early ...'

Line 80: Why does endothermy permit mammals to exploit multiple temporal niches? A little bit of logic here to go along with the citations would be helpful.

Line 84: It would help to be specific about what time frame you mean with 'period.' Is it day, year, season? All of the above (i.e., this trend occurs across temporal scales)?

Line 92: I think you are missing 'processes' from the sentence that starts with 'Bottom-up and top-down...'

Line 93-94: You have two qualifiers in this sentence when one will do. Maybe change to '...and may influence how species within an assemblage behave.'

Line 100: You get the same point across in this sentence if you remove 'were found to' and it changes the focus of this sentence from the people who did the finding to the mesopredators (which I think is a good thing).

Line 102-103: This last sentence is a really general statement that I don't necessarily agree with. Do we not know how bottom-up and top-down processes operate in nature, in general (as this

sentence suggests), or is it that we do not know how top-down or bottom-up processes shape diel activity patterns? I'm assuming here the authors mean the latter, so making this sentence be more specific would help decrease confusion to a reader.

Line 104: What questions?

Line 120: You don't need 'we predicted that' in H1.

Line 120 - 125: H1 is very long, and with all of the parentheses thrown in, I had a hard time understanding it. Maybe break into two sentences? Likewise, are all the parentheses necessary here?

Line 130 (Figure 1): The grey of the world map is very similar to those great line drawings. It ends up washing out the species up in the top left a fair bit. I would suggest making the world map darker, but that would draw more attention to that piece of background which is not ideal. This doesn't have anything to do with the information provided here (great figure)! It was just a very minor thing I noticed.

Line 135: What is meant by 'examples of species in each region?' Do these species occur across all of these regions? Is the first column for species in South America, the second column for species in Africa, etc.?

Line 137 (Figure 2): This figure is a little confusing. What is meant by the green equals signs and the orange equals signs with a strike through them? What do the colored directional arrows indicate? Maybe I am having a hard time understanding this because all hypotheses are on the same figure? It seems like there is a fair bit of white space, would it be better to have three sub-figures, one for each hypothesis?

Line 151 (Figure 3): Excellent figure, and adding in the color hue for interpolation vs extrapolation (so you can keep the same x axis across all subplots) is a brilliant idea. Can you make the axis numbers black? Also, if you wanted to declutter this plot a bit, you don't really need to label each axis on every subplot. Labeling the bottom 3 for the x axis and the left three for the y axis should make this look a little more clean.

Consistent patterns

Top-level thoughts

1. The explanations here are great, along with the probability estimates and the associated uncertainty around them. What are the results related to H2? Even if there was no support for it, putting that here (between the H1 and H3 results) would help keep the ordering the same across the different sections of the paper.

2. How variable were the results among study areas (as quantified by the random effect term in the model)? These results were not shared, and I only really discovered a random effect term was added in the 'analysis' section of the methods. Other people reading this article may be wondering whether or not random effects were included here, so it may help to give a heads up about that aspect of the model structure a little earlier in the paper.

Explanations

Top-level thoughts

1. This is a similar comment to the last section. It would help a lot with cohesion if you revisited

your hypotheses in a bit more order here (or do a bit more signposting to connect the findings here back to the different hypotheses). You already do this on line 200 for H3, which is great, but I don't see any explicit call outs to H1 or H2.

Line by line comments:

Line 196: You are missing the second 'c' for 'raccoons.'

Line 223 - 225: There was no term in the model to quantify site-specific variability (e.g., a spatial covariate) and so the model should inherently provide consistent results among sites, right? As a result, I'm not sure if this result here is remarkable, or robust. Conversely, if by 'site' you mean the 16 different 'study areas' that the cameras were deployed, then I did not see where in the results you shared that.

Line 236-239: I'm confused by this statement. The model you fit is not associated to quantifying species decline or loss, and so how would assessing variation in species diel activity patterns contradict this?

Conclusion

Line by line comments

Line 258: What is a site? Do you mean study areas? I would assume that a location a camera trap is deployed is a 'site.' This also links back to my comment on lines 223 - 225 (i.e., confusion about what a site is for this analysis).

Line 263: Missing a period to this sentence.

Methods

Top-level thoughts

1. There are few uses of 'run' which should be changed to past tense 'ran.'

2. Were continuous covariates centered and scaled?

Line by line comments

Line 290: Do you mean the `suncalc` package, not `maptools`?

Line 307-309: Did you use a random intercept model? Random slope model? Random intercept-random slope model?

Line 310: Maybe cite the Burnham & Anderson AIC book here?

Reviewer #4 (Remarks to the Author):

The paper investigates mammalian diel behavior using a large camera trapping dataset from tropical forests. The authors test if communities display diel patterns that are reflective of top-down or bottom-up forcing. The authors present a strong dataset collected across several protected areas over 9 years. The authors show that guild-level diel patterns are generally consistent across regions with large herbivores tending towards nocturnality and large predators towards diurnally, supporting that diel patterns may be determined by top-down forcing.

The introduction provides a lot of great information on the evolution of diel behavior. However, it is not clear how this research contributes to resolving any enduring questions or controversy in behavioral ecology until line 101. This knowledge gap should be in the 1st paragraph so as not to get lost and to help the reader follow the main story. The introduction can be reworked to highlight the theoretic framework and better explain how this data set provides a unique perspective.

The statistical approach taken in the paper is indirect and requires utilizing broad categories (day, night, twilight) to define activity time, losing the great temporal resolution in the camera trap data. Established methods exist to work with time data to more directly answer the hypothesis presented in the introduction. For example, to answer H1/H2 it is possible to directly compare kernel density functions of predators & prey using a Wald's test to produce an overlap estimate with corresponding error instead of the visual comparisons used. It is possible to compare species or to pool species to answer the questions in the introduction. To answer H3 Linear-circular regression can estimate how body mass relates to activity. One benefit of the author's approach is that mclogit allows random effects – but there is not a lot of evidence that non-independence is an issue here and I question if you are losing more than you gain.

This paper would also benefit from careful editing for clarity and wordiness. The author's frequent reliance on passive voice results in several unnecessarily long or convoluted sentences. For example: "For instance, in the absence of other factors, large species in warm regions may be forced to avoid overheating by avoiding activity in the hottest periods." Could be more concisely written as "Large species often avoid overheating by limiting activity to the coolest parts of the day". Also, the authors frequently use the same word twice in the same sentence.

Minor Comments:

72-78 Unclear how this section relates to the goals of the paper as it is too detailed

79 - Endothermy needs to be defined

88- Intraguild competition can also drive diel behavior, it may be worth noting this process as well. (Gutman & Dayan 2005; Sovie et al., 2019)

103 - What work is "humid" doing in this sentence? Are there arid tropical forests that may have different humidity that would change diel patterns? Also, this is a great framing for why your data set is useful but as written is too wordy – try to communicate the idea more clearly.

112 - Scansorial needs to be defined (I had to google this – how is it different from arboreal?)

198 - it is unclear if H1 or H3 is the best explanation for the behavior of large herbivores – as the model is written it is hard to compare the effect size of either factor.

302 -Why build separate model sets for each region instead of incorporating region as a fixed or random effect in the global model? Would make just as much sense to build model sets for each guild – that way the authors could incorporate a binary predator presence/absence variable for herbivores which could help tease apart which is a bigger driver of diel behavior.

Adia Sovie, PhD

Response to the comments

We are grateful for the careful and constructive reviews. We are reassured that the reviewers found the submitted manuscript of value and identified opportunities for us to further refine and improve the analyses and associated interpretations. Doing an extensive reanalysis and revising an article with so many authors it required additional time to gain the reactions, insights, and approval of everyone. We are grateful for the extension granted.

Please find our responses to the individual suggestions and comments made by the four reviewers. The major and minor changes we made will be found in the manuscript and we refer to the locations in the answers below.

Comments by reviewer 1:

Comment	Response
The authors present an extensive and interesting review of activity patterns in forest tropical mammal communities. This paper capitalises on the valuable information of community patterns obtain from camera studies and takes advantage of the extensive TEAMS camera trapping network. I found the paper interesting and think it would appeal to a general audience. I have some minor comments below	Thanks for your comments and suggestions.
1. I found the development of the hypotheses a bit confusion in parts - the use of terms top-down/bottom up are used in combination with intraguild/interguild interactions. I think using the former alone might make the arguments easier to understand. Some of these relationships may be related to dominance as well as predation. I would ask would we expect to see dominance influencing large-small herbivore interactions as well?	We revised the terms to make them clearer. We kept bottom-up and top-down processes and avoided the terms “intraguild” and “interguild” to reduce complexity. We are aware that most literature on activity patterns refers to intraguild or interguild interaction on the temporal scale, yet our study comprises a broad range of trophic groups and the use of bottom-up and top-down terms provides an elegant (and simpler) way to understand how different interactions shape the activity of a group of species. Regarding dominance among herbivores, we did not establish a hypothesis on intraguild avoidance. Most studies on activity have assessed one-to-one species, more commonly assessing prey-predator interactions or species of the same or similar size.
2. I would think we might expect some of these hypotheses more likely than others. The pred-prey interactions is complicated as both predators and prey may be co-evolving strategies whereas, large predators may not be trying to map activities based on small predators so there is the clear opportunity for a joint strategy and more likely clearer results. Also this could then lead to clearer strategies of small prey avoidance.. but then this is difficult to separate from temperature regulation of small species. I wonder if the authors could suggest if one hypothesis is dominant and likely to then lead to further activity patterns in these communities.	We now have addressed the three hypotheses separately and found evidence for all of them. Briefly, it is likely, that the activity of herbivores and insectivores in two regions (Afrotropics and Indo-Malayan tropics) are mostly constrained by thermoregulatory limits. We found evidence for a bottom-up process shaping the activity of large predators. And the evidence for top-down when we analysed the activity of small omnivores and insectivores suggests avoidance of large predators. Note changes in the introduction lines 138-147, methods lines: 399-416, results lines: 194-216, and Fig. 4. Nevertheless, we could not test which of the three hypothesis prevails within a joint model because two variables, temperature and hours of the day are correlated (e.g., the temperature increases after

	sunrise and reaches its maximum around noon). Thus, fitting a model with these variables could lead to biased results.
3. Labelling could be better in Fig 2.. can you put the H1 H2 etc on the plot? Avoid too many terms? Would you expect any interaction between small and large herbivores?	Thanks for your suggestion, we addressed your comment and simplified the figure in the document. Regarding the interactions on the temporal scale between small and large herbivores, as mentioned before, we did not test these interactions because there is no evidence in the literature for a temporal interaction among herbivores of different sizes. There are only reports of segregation in activity patterns in herbivores with similar body sizes¹.
4. Fig 3. It might be nice to see individual activity charts for some of the species marked on the plots here. Perhaps in a separate panel to show how the predicted and observed ratios of activity compare.	Thanks for your recommendation. We included the predictions and observed ratios in the Supplementary material Fig. S4. Also, examples of the activity of some species in Fig. S3. Including a panel with observed ratios makes the figure too complex.

Comments by reviewer 2:

Comment	Response
The authors report the quantification of diel organization of different tropical forest species guilds using data from standardized camera trapping protocols in 16 protected areas across three continents. The outcome is a remarkable consistency in the size-dependent temporal organization of daily activity in different trophic guilds. The authors pose three hypotheses on the variance in diel organization within and between these guilds and argue that trophic interactions provide the best explanation to support these. The manuscript is clear and well written; a pleasure to read. I have only minor comments. Although the observation of homology in daily organization of activity in guilds across tropical rainforests is highly interesting and a strong contribution in itself, the explanation why this actually happens is more speculative. The authors argue this depends on generalized inter- and intra-guild trophic interactions. This reasoning potentially benefits from a quantification of variance in diel organization within (size classes of) guilds. Furthermore, the study strongly hinges on the assumption that larger species are always preying on smaller species, which is of course logical, but predator-prey interaction may not always depend on predator-prey size differences. Anyway, possible exceptions(?) in top-down interactions do not blur the size-driven diel activity organization. If variation in size-dependent predator-prey interaction can be quantified, this would be good. Anyway, putting these hypotheses forward is useful in itself.	Thank you for your comments. We now tested quantitatively the influence of the activity of predators on the activity of prey to avoid speculation of our findings and we added additional descriptions in the introduction lines 138-147, methods lines: 399-416, and results lines: 194-216 and Fig. 4.
Insectivores, line 250-252 – I assume the variation in temporal organization of insect prey is huge anyway, and cascades to the insectivore level?	That is a good point. We now included a sentence mentioning the possible cascading effects line 281-283. However, we did not find literature reporting overlap with the activity of insects. Some insectivore species find their prey by a developed sensorial system as olfaction², and can find their prey when they are inactive. For example, aardvarks forages when their prey species are inactive and are concentrated in the colonies or nests, to acquire food efficiently³. Yet, we do not know whether the activity of insect prey determine the activity of insectivore mammals.
Line 313 – what is meant with independent events here?	We have now made a clarification in the explanation at line 362: “time interval between pictures > 1-hour per camera for a given species”

Comments by reviewer 3:

Comments	Response
In this paper the authors assessed diel activity patterns of mammals in humid tropical forests across three biogeographical regions. I was very excited to get to read this manuscript, as it aligns with my own research interests. Overall, this is a high quality piece of research that the authors should be proud of. I also commend the authors for using a multinomial model to quantify diel activity patterns, as it gave them the ability to more directly assess their hypotheses in relation to body size. I only have what I would consider minor suggestions to this article, which hopefully will improve some parts that were a little confusing to me and improve the cohesion of this piece of research. If you have any specific questions about my comments, feel free to reach out to me directly (Mason Fidino, mfidino@lpzoo.org). Following these two 'top-level' comments, I have smaller comments about various sections of the manuscript. I hope the authors find these suggestions helpful.	Thank you for the review and positive comments. They have helped us to improve this manuscript. We appreciate your offer to help us with the analysis. For this study we wish to limit our evaluation to test the hypothesis proposed which we feel is enough for one, already ambitious, article, and we will gather the additional questions and hypotheses for future and separate analyses.
1. On the use of the phrase daily activity patterns versus diel activity pattern. Since the authors are using both terms, and they are effectively synonymous, I suggest that they just try to stay consistent throughout the text. My personal preference is for diel activity, as I feel it is a bit more specific, but the authors are free to choose whatever phrasing they like.	We have revised and used diel in the revised manuscript. The changes are found in the manuscript.
2. I am a big fan of using a multinomial model for diel activity patterns, as it allows you to incorporate continuous covariates to look at changes in diel activity patterns. However, one unique aspect of applying such a model to these data is that the temporal 'bins' for each time period are different amounts of time. What is great though, is that we know how much time is available for each diel period over 24 hours. If you wanted to convert your model into more of a temporal resource selection function, so you are quantifying use relative to availability, you could add a log offset term to the model for each diel period that is the amount of hours associated to each diel category. You should end up with essentially the same results (i.e., the slope terms should not change, only the intercepts). Right now the intercepts are a mix of use & availability, which is not ideal. See Gallo et al. (2022) for an example of a multinomial model with a log offset term. Gallo, T., Fidino, M., Gerber, B., Ahlers, A. A., Angstmann, J. L., Amaya, M., ... & Magle, S. (2021). Mammals adjust diel activity across gradients of urbanization. bioRxiv. Link to preprint: https://www.biorxiv.org/content/10.1101/2021.09.24.461702v1.abstract	Thank you very much for your suggestion. We will not attempt this here (see comments above) but agree that the issues are worthy of exploration. We are planning to use the models suggested in our next paper. For this manuscript, we focused on evidence of mutual influences between species and the effect of body size employing a set of analyses (GLMM) to strengthen hypothesis testing and the statistical part in general.

Summary: Top-level thoughts	
1. I like that the abstract provides some direction of certain relationships the authors found (e.g., larger herbivores tended to be more nocturnal). If space allows, it would be great to add in some small info about effect size and the like here (e.g., larger herbivores were X times more likely to be nocturnal than smaller herbivores).	Agree. We are out of space in the Abstract, but now mention the quantitative difference in the main text. Line 169, 173
Summary: Line by line comments	
Line 47: Very minor thing, but it reads a little weird to use 'specific' twice in the same sentence like this. Is the second 'specific' really needed?	Thanks. It has been incorporated
Line 48: It takes a second to connect 'these patterns' to what I assume you mean 'daily activity patterns'? May help to just exchange 'these patterns' for 'diel activity patterns.'	Thanks. It has been incorporated
Introduction: Line by line comments	
Line 71: There are a lot of behaviours brought up in this paragraph, and so I'm not certain what 'such behaviours' refers to for mammals here. Also, adding mammals at the very end of this paragraph comes a bit out of left field. I think it may read better if you had a different crystallizing point at the end of this paragraph and then start the following paragraph with something like 'Mammals illustrate a broad range of diel activity patterns and occupy all temporal niches (day, night, twilight). However, early ...'	Thanks for the recommendation. We took into consideration your comments and comments from reviewer 4 to revise and clarify this paragraph. Line 71-85
Line 80: Why does endothermy permit mammals to exploit multiple temporal niches? A little bit of logic here to go along with the citations would be helpful.	Thanks. It has been incorporated
Line 84: It would help to be specific about what time frame you mean with 'period.' Is it day, year, season? All of the above (i.e., this trend occurs across temporal scales)?	Agree. We focused on the daily temperature changes, and we mentioned "period of the day". Note the changes in line 82 .
Line 92: I think you are missing 'processes' from the sentence that starts with 'Bottom-up and top-down...'	Agree. It has been incorporated in line 91
Line 93-94: You have two qualifiers in this sentence when one will do. Maybe change to '...and may influence how species within an assemblage behave.'	We expanded this paragraph to improve its understanding. Lines 87:105
Line 100: You get the same point across in this sentence if you remove 'were found to' and it changes the focus of this sentence from the people who did the finding to the mesopredators (which I think is a good thing).	Thanks. It has been incorporated
Line 102-103: This last sentence is a really general statement that I don't necessarily agree with. Do we not know how bottom-up and top-down processes operate in nature, in general (as this sentence suggests), or is it that we do not know how top-down or bottom-up processes shape diel activity patterns? I'm assuming here the authors mean the latter, so making this sentence be more specific would help decrease confusion to a reader.	Thanks, we edited this sentence to clarify.
Line 104: What questions?	Good point. We rewrote the first sentence of this paragraph. Note changes in line 106-107

Line 120: You don't need 'we predicted that' in H1.	Thanks. It has been addressed
Line 120 - 125: H1 is very long, and with all of the parentheses thrown in, I had a hard time understanding it. Maybe break into two sentences? Likewise, are all the parentheses necessary here?	Agree. We changed the order of the hypothesis and shortened them to make it clear and we decrease the use of parentheses.
Line 130 (Figure 1): The grey of the world map is very similar to those great line drawings. It ends up washing out the species up in the top left a fair bit. I would suggest making the world map darker, but that would draw more attention to that piece of background which is not ideal. This doesn't have anything to do with the information provided here (great figure)! It was just a very minor thing I noticed.	Thanks. We followed your suggestion. We plotted a new figure (Fig. R1). However, the map in the background draws too much attention, thus we considered to keep the figure from our first version in the manuscript.
Line 135: What is meant by 'examples of species in each region?' Do these species occur across all of these regions? Is the first column for species in South America, the second column for species in Africa, etc.?	Yes, species in each column are present in the respective region. Note that this is now revised and clarified in the legend of the figure 1.
Line 137 (Figure 2): This figure is a little confusing. What is meant by the green equals signs and the orange equals signs with a strike through them? What do the colored directional arrows indicate? Maybe I am having a hard time understanding this because all hypotheses are on the same figure? It seems like there is a fair bit of white space, would it be better to have three sub-figures, one for each hypothesis?	Ok, thanks. We have changed the figure into three sections to improve its understanding.
Line 151 (Figure 3): Excellent figure, and adding in the color hue for interpolation vs extrapolation (so you can keep the same x axis across all subplots) is a brilliant idea. Can you make the axis numbers black? Also, if you wanted to declutter this plot a bit, you don't really need to label each axis on every subplot. Labeling the bottom 3 for the x axis and the left three for the y axis should make this look a little more clean.	Thanks for the recommendation. We plotted a new figure; it has common axis labels, and the axis text is black as suggested.
Consistent patterns: Top-level thoughts	
1. The explanations here are great, along with the probability estimates and the associated uncertainty around them. What are the results related to H2? Even if there was no support for it, putting that here (between the H1 and H3 results) would help keep the ordering the same across the different sections of the paper.	Thanks for the guidance. We reorganized the results and discussion as suggested.
2. How variable were the results among study areas (as quantified by the random effect term in the model)? These results were not shared, and I only really discovered a random effect term was added in the 'analysis' section of the methods. Other people reading this article may be wondering whether or not random effects were included here, so it may help to give a heads up about that aspect of the model structure a little earlier in the paper.	We added "random intercept" in the introduction line 132 . In Table S3 we included the estimate for each region and the variance-covariance matrix indicating the estimate and standard error from the random effects.
Explanations: Top-level thoughts	
1. This is a similar comment to the last section. It would help a lot with cohesion if you revisited your hypotheses in a bit more order here (or do a bit more signposting to connect the findings here back to the different hypotheses). You already do this on line 200 for H3, which is great, but I don't see any explicit call outs to H1 or H2.	Thanks, yes, we agree. As mentioned before, we changed the structure of the results and discussion to improve the order and connection with our hypotheses.

Explanations: Line by line comments:	
Line 196: You are missing the second 'c' for 'raccoons.'	Oops, thanks. Incorporated in the text.
Line 223 - 225: There was no term in the model to quantify site-specific variability (e.g., a spatial covariate) and so the model should inherently provide consistent results among sites, right? As a result, I'm not sure if this result here is remarkable, or robust. Conversely, if by 'site' you mean the 16 different 'study areas' that the cameras were deployed, then I did not see where in the results you shared that.	Thanks for your comment, as you pointed out, the results are consistent across regions. Study areas (i.e., protected areas) were included as a random intercept within each region. We clarified this in the text (e.g., line 132)
Line 236-239: I'm confused by this statement. The model you fit is not associated to quantifying species decline or loss, and so how would assessing variation in species diel activity patterns contradict this?	Thanks, we agree, we have removed this paragraph.
Conclusion: Line by line comments	
Line 258: What is a site? Do you mean study areas? I would assume that a location a camera trap is deployed is a 'site.' This also links back to my comment on lines 223 - 225 (i.e., confusion about what a site is for this analysis).	Thanks. We changed to "study area" (i.e., protected area)
Line 263: Missing a period to this sentence.	Thanks. It has been incorporated.
Methods: Top-level thoughts	
1. There are few uses of 'run' which should be changed to past tense 'ran.'	Thanks. It has been incorporated.
2. Were continuous covariates centered and scaled?	We now have added the type of variables (discrete and continuous) in the text Line 377 . Body mass was the only continuous variable and was log-transformed, instead of scaled.
Methods: Line by line comments	
Line 290: Do you mean the `suncalc` package, not `maptools`?	We added the functions we employed from the package 'maptools' ('crepuscule' and 'sunrise') in the text to clarify how we extracted the hours of sunrise, sunset and nautical dawn and dusk for a specific location and date. Line 366-369 .
Line 307-309: Did you use a random intercept model? Random slope model? Random intercept-random slope model?	We employed a random intercept model. We added it in the introduction in line 132 . We tried to run models with random slope, but the models did not converge.
Line 310: Maybe cite the Burnham & Anderson AIC book here?	Agree, now we incorporated it. Line 393 .

Comments by reviewer 4:

The paper investigates mammalian diel behavior using a large camera trapping dataset from tropical forests. The authors test if communities display diel patterns that are reflective of top-down or bottom-up forcing. The authors present a strong dataset collected across several protected areas over 9 years. The authors show that guild-level diel patterns are generally consistent across regions with large herbivores tending towards nocturnality and large predators towards diurnally, supporting that diel patterns may be determined by top-down forcing.	Many thanks. Your comments and suggestions helped us improve the paper.
The introduction provides a lot of great information on the evolution of diel behavior. However, it is not clear how this research contributes to resolving any enduring questions or controversy in behavioral ecology until line 101. This knowledge gap should be in the 1st paragraph so as not to get lost and to help the reader follow the main story. The introduction can be reworked to highlight the theoretic framework and better explain how this data set provides a unique perspective.	Thanks for your comment, we have made important changes in the introduction to address these concerns. We moved the knowledge gap to lines 71-72.
The statistical approach taken in the paper is indirect and requires utilizing broad categories (day, night, twilight) to define activity time, losing the great temporal resolution in the camera trap data. Established methods exist to work with time data to more directly answer the hypothesis presented in the introduction. For example, to answer H1/H2 it is possible to directly compare kernel density functions of predators & prey using a Wald's test to produce an overlap estimate with corresponding error instead of the visual comparisons used. It is possible to compare species or to pool species to answer the questions in the introduction.	We acknowledge that employing methods to more formally test activity allows us to assess our hypothesis regarding top-down or bottom-up processes and followed the suggestion of the reviewer. We now estimated the overlap and test the differences between groups of species for each protected area to exemplify our results. A description of the methodological approach is provided in lines 417-428, and the kernel density plots are included in the Supplementary Material (PDF). Nevertheless, to make an appropriate inference of general patterns across regions, we would need to pool the data. We found that Ridout & Linkie (2009)⁴ in their paper stated that: "...the values of overlap obtained for the data pooled across areas will exceed the average value for the separate areas..., so that pooling the data across sites substantially overestimates the extent of overlap within areas." Thus, we considered that pooling data would ignore important variation and potentially mask effects or even lead to biases. To overcome this issue, we added a new analysis employing GLMMs to assess if there were positive or negative effects on the activity of pairwise comparison of trophic guilds and sizes employing the number of events on an hourly basis. This analysis allowed us to include random effects because it is important to account for the variation in sampling effort among protected areas. We also ran a version of this analysis with the estimated values from the fitted activity models (kernel density) from the overlap package. Qualitatively, those results were similar. However, we preferred to use the results based on hourly data, as it

	integrates additional temporal variation and allows it to propagate the uncertainty estimates around the coefficients.
To answer H3 Linear-circular regression can estimate how body mass relates to activity. One benefit of the author's approach is that mclogit allows random effects – but there is not a lot of evidence that non-independence is an issue here and I question if you are losing more than you gain.	We now fitted (and added here) a linear-circular regression with the function “linearKern” from the package “activity” and found support for the relationship between activity and body mass Fig. R2. For instance, large herbivores are more likely active at night, while large carnivores and omnivores appeared to be active more likely during the day. These results were consistent with our predictions from the multinomial model (Fig. 1 in the manuscript). Therefore, we kept the results from the multinomial analysis as we consider it important to keep the random effect, and it helped us to show the results visually neat for a broad audience. Our analysis allowed us to test the relationship between the activity of trophic guilds and body mass in all regions. This was not the case with the linear-circular regression, we could not assess the relationship between body mass and hours of the day for omnivores and insectivores in the Afrotropics.
This paper would also benefit from careful editing for clarity and wordiness. The author's frequent reliance on passive voice results in several unnecessarily long or convoluted sentences. For example: “For instance, in the absence of other factors, large species in warm regions may be forced to avoid overheating by avoiding activity in the hottest periods.” Could be more concisely written as “Large species often avoid overheating by limiting activity to the coolest parts of the day”. Also, the authors frequently use the same word twice in the same sentence.	Thanks for your comment; we checked and edited the manuscript to be more precise.
Minor Comments:	
2-78 Unclear how this section relates to the goals of the paper as it is too detailed	Thanks, we agree. We have now summarised this section to make it clear for the reader.
79 - Endothermy needs to be defined	Thanks. We have now included the concept in parenthesis. Line 75“(i.e., generation and regulation of body temperature)”
88- Intraguild competition can also drive diel behavior, it may be worth noting this process as well. (Gutman & Dayan 2005; Sovie et al., 2019)	We agree. We now included one reference Line 86 (Sovie et al., 2019)
103 – What work is “humid” doing in this sentence? Are there arid tropical forests that may have different humidity that would change diel patterns? Also, this is a great framing for why your data set is useful but as written is too wordy – try to communicate the idea more clearly.	Good point. We have edited this paragraph to make our framing clear. Line 106 – 110.
112 - Scansorial needs to be defined (I had to google this – how is it different from arboreal?)	Thanks, we agree. It has been incorporated. Line 114-115
198 - it is unclear if H1 or H3 is the best explanation for the behavior of large herbivores – as the model is written it is hard to compare the effect size of either factor.	Thank you for pointing this out. In this revised version of our manuscript, as mentioned before we have added new analyses. We tested our hypothesis with two main analyses (GLMM and overlap analyses for each pairwise comparison). Despite not finding a way to test which factor is stronger in the activity of tropical mammals, on one hand,

	our findings suggest that the activity of herbivores and insectivores is mainly shaped by the thermoregulatory constraints. On the other hand, since the activity of herbivores was less likely shaped by the activity of carnivores, this suggests that predation risk is not strong enough to cascade and shift the activity of herbivores. Yet, the activity of other guilds as small carnivores, omnivores, and small insectivores showed a negative effect when compared with the activity of large carnivores.
302 -Why build separate model sets for each region instead of incorporating region as a fixed or random effect in the global model? Would make just as much sense to build model sets for each guild – that way the authors could incorporate a binary predator presence/absence variable for herbivores which could help tease apart which is a bigger driver of diel behavior.	Thanks for your helpful suggestion it prompted us to build the models and test the hypothesis. As mentioned, we employed GLMM to assess how the activity of prey (e.g., large herbivores) is related to the activity of predators (e.g., large carnivores). We employed the number of independent events for each group and include the protected areas as a random intercept. Both type of models, the multinomial models and the new added GLMMs were built separately because we aimed to confirm if there is consistency or divergency across biogeographic regions.

Figure 1R. Optional figure for Figure 3.

Figure R2. Linear circular relationship of body mass and activity at the regional level. If the fitted line lays outside the confidence values (grey area) of a null hypothesis, the alternative hypothesis is accepted. We found that in general, large carnivores and omnivores are more likely active during hours of the day, and we find the opposite for herbivores. Larger species active most likely during the night. We found that insectivores are inconsistent, in the Neotropics large species were more active during the day, while in the Indo-Malayan tropics large species tended to be nocturnal. It was insufficient observation to test with this framework the relationship of body mass with the activity of omnivores and insectivores in the Afrotropics.

References:

- 1 de Oliveira, M. L. *et al.* Phylogenetic signal in the circadian rhythm of morphologically convergent species of Neotropical deer. *Mammalian Biology* **81**, 281-289 (2016).
- 2 McAdam, D. W. & Way, J. S. Olfactory discrimination in the giant anteater. *Nature* **214**, 316-317 (1967).
- 3 Taylor, W. & Skinner, J. Adaptations of the armadillo for survival in the Karoo: a review. *Transactions of the Royal Society of South Africa* **59**, 105-108 (2004).

- 4 Ridout, M. S. & Linkie, M. Estimating overlap of daily activity patterns from camera trap data. *Journal of Agricultural, Biological, and Environmental Statistics* **14**, 322-337 (2009).

Reviewer comments, second round –

Reviewer #1 (Remarks to the Author):

The authors present an extensive and interesting review of activity patterns in forest tropical mammal communities. This paper capitalises on the valuable information of community patterns obtain from camera studies and takes advantage of the extensive TEAMS camera trapping network.

I found the paper interesting and think it would appeal to a general audience. I have some minor comments below:

1. I found the development of the hypotheses a bit confusion in parts - the use of terms top-down/bottom up are used in combination with intraguild/interguild interactions. I think using the former alone might make the arguments easier to understand. Some of these relationships may be related to dominance as well as predation. I would ask would we expect to see dominance influencing large-small herbivore interactions as well?
2. I would think we might expect some of these hypotheses more likely than others. The pred-prey intereactions is complicated as both predators and prey may be co-evolving strategies whereas, large predators may not be trying to map activities based on small predators so there is the clear opportunity for a joint strategy and more likely clearer results. Also this could then lead to clearerr strategies of small prey avoidance.. but then this is difficult to separate from temperature regulation of small species. I wonder if the authors could suggest if one hypothesis is dominant and likely to then lead to further activity patterns in these communities.
3. Labelling could be better in Fig 2.. can you put the H1 H2 etc on the plot? Avoid too many terms? Would you expect any interaction between small and large herbivores?
4. Fig 3. It might be nice to see individual activity charts for some of the species marked on the plots here. Perhaps in a separate panel to show how the predicted and observed ratios of activity compare.

Reviewer #2 (Remarks to the Author):

Reviewer comments to the manuscript "Consistent daily activity patterns across tropical forest mammal communities" by Vallejo-Vargas et al.

The authors report the quantification of diel organization of different tropical forest species guilds using data from standardized camera trapping protocols in 16 protected areas across three continents. The outcome is a remarkable consistency in the size-dependent temporal organization of daily activity in different trophic guilds. The authors pose three hypotheses on the variance in diel organization within and between these guilds and argue that trophic interactions provide the best explanation to support these. The manuscript is clear and well written; a pleasure to read. I have only minor comments.

Although the observation of homology in daily organization of activity in guilds across tropical rainforests is highly interesting and a strong contribution in itself, the explanation why this actually happens is more speculative. The authors argue this depends on generalized inter- and intra-guild trophic interactions. This reasoning potentially benefits from a quantification of variance in diel organization within (size classes of) guilds. Furthermore, the study strongly hinges on the assumption that larger species are always preying on smaller species, which is of course logical, but predator-prey interaction may not always depend on predator-prey size differences. Anyway, possible exceptions(?) in top-down interactions do not blur the size-driven diel activity organization. If variation in size-dependent predator-prey interaction can be quantified, this would be good. Anyway, putting these hypotheses forward is useful in itself.

Insectivores, line 250-252 – I assume the variation in temporal organization of insect prey is huge anyway, and cascades to the insectivore level?

Line 313 – what is meant with independent events here?

Reviewer #3 (Remarks to the Author):

Review for:

Consistent daily activity patterns across tropical forest mammal communities

Note: This review is written in markdown format. I've also attached a pdf if that is easier to read.

In this paper the authors assessed diel activity patterns of mammals in humid tropical forests across three biogeographical regions. I was very excited to get to read this manuscript, as it aligns with my own research interests.

Overall, this is a high quality piece of research that the authors should be proud of. I also commend the authors for using a multinomial model to quantify diel activity patterns, as it gave them the ability to more directly assess their hypotheses in relation to body size.

I only have what I would consider minor suggestions to this article, which hopefully will improve some parts that were a little confusing to me and improve the cohesion of this piece of research. If you have any specific questions about my comments, feel free to reach out to me directly (Mason Fidino, mfidino@lpzoo.org).

Following these two 'top-level' comments, I have smaller comments about various sections of the manuscript. I hope the authors find these suggestions helpful.

1. On the use of the phrase daily activity patterns versus diel activity pattern. Since the authors are using both terms, and they are effectively synonymous, I suggest that they just try to stay consistent throughout the text. My personal preference is for diel activity, as I feel it is a bit more specific, but the authors are free to choose whatever phrasing they like.

2. I am a big fan of using a multinomial model for diel activity patterns, as it allows you to incorporate continuous covariates to look at changes in diel activity patterns. However, one unique aspect of applying such a model to these data is that the temporal 'bins' for each time period are different amounts of time. What is great though, is that we know how much time is available for each diel period over 24 hours. If you wanted to convert your model into more of a temporal resource selection function, so you are quantifying use relative to availability, you could add a log offset term to the model for each diel period that is the amount of hours associated to each diel category. You should end up with essentially the same results (i.e., the slope terms should not change, only the intercepts). Right now the intercepts are a mix of use & availability, which is not ideal. See Gallo et al. (2022) for an example of a multinomial model with a log offset term.

...

Gallo, T., Fidino, M., Gerber, B., Ahlers, A. A., Angstmann, J. L., Amaya, M., ... & Magle, S. (2021). Mammals adjust diel activity across gradients of urbanization. bioRxiv.

...

Link to preprint:

<https://www.biorxiv.org/content/10.1101/2021.09.24.461702v1.abstract>

Summary

Top-level thoughts

1. I like that the abstract provides some direction of certain relationships the authors found (e.g., larger herbivores tended to be more nocturnal). If space allows, it would be great to add in some small info about effect size and the like here (e.g., larger herbivores were X times more likely to be nocturnal than smaller herbivores).

Line by line comments

Line 47: Very minor thing, but it reads a little weird to use 'specific' twice in the same sentence like this. Is the second 'specific' really needed?

Line 48: It takes a second to connect 'these patterns' to what I assume you mean 'daily activity patterns'? May help to just exchange 'these patterns' for 'diel activity patterns.'

Introduction

Line by line comments

Line 71: There are a lot of behaviours brought up in this paragraph, and so I'm not certain what 'such behaviours' refers to for mammals here. Also, adding mammals at the very end of this paragraph comes a bit out of left field. I think it may read better if you had a different crystallizing point at the end of this paragraph and then start the following paragraph with something like 'Mammals illustrate a broad range of diel activity patterns and occupy all temporal niches (day, night, twilight). However, early ...'

Line 80: Why does endothermy permit mammals to exploit multiple temporal niches? A little bit of logic here to go along with the citations would be helpful.

Line 84: It would help to be specific about what time frame you mean with 'period.' Is it day, year, season? All of the above (i.e., this trend occurs across temporal scales)?

Line 92: I think you are missing 'processes' from the sentence that starts with 'Bottom-up and top-down...'

Line 93-94: You have two qualifiers in this sentence when one will do. Maybe change to '...and may influence how species within an assemblage behave.'

Line 100: You get the same point across in this sentence if you remove 'were found to' and it changes the focus of this sentence from the people who did the finding to the mesopredators (which I think is a good thing).

Line 102-103: This last sentence is a really general statement that I don't necessarily agree with. Do we not know how bottom-up and top-down processes operate in nature, in general (as this

sentence suggests), or is it that we do not know how top-down or bottom-up processes shape diel activity patterns? I'm assuming here the authors mean the latter, so making this sentence be more specific would help decrease confusion to a reader.

Line 104: What questions?

Line 120: You don't need 'we predicted that' in H1.

Line 120 - 125: H1 is very long, and with all of the parentheses thrown in, I had a hard time understanding it. Maybe break into two sentences? Likewise, are all the parentheses necessary here?

Line 130 (Figure 1): The grey of the world map is very similar to those great line drawings. It ends up washing out the species up in the top left a fair bit. I would suggest making the world map darker, but that would draw more attention to that piece of background which is not ideal. This doesn't have anything to do with the information provided here (great figure)! It was just a very minor thing I noticed.

Line 135: What is meant by 'examples of species in each region?' Do these species occur across all of these regions? Is the first column for species in South America, the second column for species in Africa, etc.?

Line 137 (Figure 2): This figure is a little confusing. What is meant by the green equals signs and the orange equals signs with a strike through them? What do the colored directional arrows indicate? Maybe I am having a hard time understanding this because all hypotheses are on the same figure? It seems like there is a fair bit of white space, would it be better to have three sub-figures, one for each hypothesis?

Line 151 (Figure 3): Excellent figure, and adding in the color hue for interpolation vs extrapolation (so you can keep the same x axis across all subplots) is a brilliant idea. Can you make the axis numbers black? Also, if you wanted to declutter this plot a bit, you don't really need to label each axis on every subplot. Labeling the bottom 3 for the x axis and the left three for the y axis should make this look a little more clean.

Consistent patterns

Top-level thoughts

1. The explanations here are great, along with the probability estimates and the associated uncertainty around them. What are the results related to H2? Even if there was no support for it, putting that here (between the H1 and H3 results) would help keep the ordering the same across the different sections of the paper.

2. How variable were the results among study areas (as quantified by the random effect term in the model)? These results were not shared, and I only really discovered a random effect term was added in the 'analysis' section of the methods. Other people reading this article may be wondering whether or not random effects were included here, so it may help to give a heads up about that aspect of the model structure a little earlier in the paper.

Explanations

Top-level thoughts

1. This is a similar comment to the last section. It would help a lot with cohesion if you revisited

your hypotheses in a bit more order here (or do a bit more signposting to connect the findings here back to the different hypotheses). You already do this on line 200 for H3, which is great, but I don't see any explicit call outs to H1 or H2.

Line by line comments:

Line 196: You are missing the second 'c' for 'raccoons.'

Line 223 - 225: There was no term in the model to quantify site-specific variability (e.g., a spatial covariate) and so the model should inherently provide consistent results among sites, right? As a result, I'm not sure if this result here is remarkable, or robust. Conversely, if by 'site' you mean the 16 different 'study areas' that the cameras were deployed, then I did not see where in the results you shared that.

Line 236-239: I'm confused by this statement. The model you fit is not associated to quantifying species decline or loss, and so how would assessing variation in species diel activity patterns contradict this?

Conclusion

Line by line comments

Line 258: What is a site? Do you mean study areas? I would assume that a location a camera trap is deployed is a 'site.' This also links back to my comment on lines 223 - 225 (i.e., confusion about what a site is for this analysis).

Line 263: Missing a period to this sentence.

Methods

Top-level thoughts

1. There are few uses of 'run' which should be changed to past tense 'ran.'

2. Were continuous covariates centered and scaled?

Line by line comments

Line 290: Do you mean the `suncalc` package, not `maptools`?

Line 307-309: Did you use a random intercept model? Random slope model? Random intercept-random slope model?

Line 310: Maybe cite the Burnham & Anderson AIC book here?

Reviewer #4 (Remarks to the Author):

The paper investigates mammalian diel behavior using a large camera trapping dataset from tropical forests. The authors test if communities display diel patterns that are reflective of top-down or bottom-up forcing. The authors present a strong dataset collected across several protected areas over 9 years. The authors show that guild-level diel patterns are generally consistent across regions with large herbivores tending towards nocturnality and large predators towards diurnally, supporting that diel patterns may be determined by top-down forcing.

The introduction provides a lot of great information on the evolution of diel behavior. However, it is not clear how this research contributes to resolving any enduring questions or controversy in behavioral ecology until line 101. This knowledge gap should be in the 1st paragraph so as not to get lost and to help the reader follow the main story. The introduction can be reworked to highlight the theoretic framework and better explain how this data set provides a unique perspective.

The statistical approach taken in the paper is indirect and requires utilizing broad categories (day, night, twilight) to define activity time, losing the great temporal resolution in the camera trap data. Established methods exist to work with time data to more directly answer the hypothesis presented in the introduction. For example, to answer H1/H2 it is possible to directly compare kernel density functions of predators & prey using a Wald's test to produce an overlap estimate with corresponding error instead of the visual comparisons used. It is possible to compare species or to pool species to answer the questions in the introduction. To answer H3 Linear-circular regression can estimate how body mass relates to activity. One benefit of the author's approach is that mclogit allows random effects – but there is not a lot of evidence that non-independence is an issue here and I question if you are losing more than you gain.

This paper would also benefit from careful editing for clarity and wordiness. The author's frequent reliance on passive voice results in several unnecessarily long or convoluted sentences. For example: "For instance, in the absence of other factors, large species in warm regions may be forced to avoid overheating by avoiding activity in the hottest periods." Could be more concisely written as "Large species often avoid overheating by limiting activity to the coolest parts of the day". Also, the authors frequently use the same word twice in the same sentence.

Minor Comments:

72-78 Unclear how this section relates to the goals of the paper as it is too detailed

79 - Endothermy needs to be defined

88- Intraguild competition can also drive diel behavior, it may be worth noting this process as well. (Gutman & Dayan 2005; Sovie et al., 2019)

103 - What work is "humid" doing in this sentence? Are there arid tropical forests that may have different humidity that would change diel patterns? Also, this is a great framing for why your data set is useful but as written is too wordy – try to communicate the idea more clearly.

112 - Scansorial needs to be defined (I had to google this – how is it different from arboreal?)

198 - it is unclear if H1 or H3 is the best explanation for the behavior of large herbivores – as the model is written it is hard to compare the effect size of either factor.

302 -Why build separate model sets for each region instead of incorporating region as a fixed or random effect in the global model? Would make just as much sense to build model sets for each guild – that way the authors could incorporate a binary predator presence/absence variable for herbivores which could help tease apart which is a bigger driver of diel behavior.

Adia Sovie, PhD

Response to the comments

We are thankful for the positive and careful reviews to our manuscript. Please find here our responses to each suggestion and comment made by the reviewers. All changes we made will be found in the manuscript and we refer to the locations in the answers below.

Reviewer #2

Reviewer comments to the manuscript ‘Consistent diel activity patterns of forest mammals across tropical regions’ by Vallejo-Vargas et al. I think the manuscript has improved substantially by reviewer comments, apart from the fact that the ms was, to my opinion, already in a pretty good shape during the First review round. I continue to like and appreciate the work, and I think the paper greatly contributes to our understanding of how activity patterns are shaped. I have no further questions or comments at this point. Looking forward to see it published.	Thanks for your positive feedback, we really appreciate it.
--	--

Reviewer #3:

I was one of the previous reviewers of this manuscript. I liked the first submission a lot, and the revisions that the authors did here are great. My only very minor sticking point is some of the heavy reliance on R functions to explain some of their methodology (see below). Great work on this paper, I look forward to citing it! - Mason Fidino	Thank you for your comments and suggestions that helped us to improve the manuscript.
## Results --- ### Line by line comments Line 197 - 2015: This may just be due to the way the pdf of the MS got spun up, but it seems like the symbols for each parameter in these paragraphs are missing (i.e., there is nothing to the left of =, it is just a blank space).	Thanks for pointing this out. The symbol is \square for the coefficients. It might have disappeared when it was converted to a PDF in the platform. We will make sure they will appear in the final version.
## Methods ### Top-level thoughts 1. One aspect of the methods that I do not agree with is the heavy reliance on using R functions to explain the methodology. Functions change over time, packages become deprecated or abandoned, and we may be using some other programming language than R in 20 years. This was most evident with the overlap package (the multinomial modeling was explained just fine). This paper (open access) has some great suggestions related to this: https://besjournals.onlinelibrary.wiley.com/doi/full/10.1111/2041-210X.13105	Thanks for your comment. We agree, and in the new version we explained the statistical approaches employed and excluded the functions. Lines 402-410
### Line by line comments Line 393: typo	We corrected it.

## Tables & figures --- ### Top-level thoughts Figure 2: Those + and - signs on the left could be a little bit bigger. I fear they will disappear if the figure is made any smaller. Great edit on this figure though, the hypotheses are much more clear!	Good recommendation. We increased the font size and symbols in case they shrink in the publication.
--	--

Reviewer #4:

This paper presents a strong data set to investigate large scale drivers of diel behavior in mammals. There are still some lingering concerns from the 1st round of reviews. My primary concern with utilizing broad categories to describe animal behavior was not adequately addressed within the text of the manuscript. In their response to reviewers the authors provide reasonable arguments to defend their position, which should be incorporated into the main text. The overlap estimates they did include were not well integrated into the paper and should be dropped unless they are better utilized to address the driving questions of the paper. Also, the writing remains very rough, with many typos, errors, and wordy sections.	Dear reviewer, we appreciate the detailed feedback. We incorporated the results and the arguments provided in the last response into our manuscript Lines 152-156, 199-206, 297-304. In addition, the native English speakers among our co-authors have given the manuscript another careful read to collect errors and make the text more concise.
line by line comments:	
Abstract: Needs a concluding/ summary sentence	Thanks, we included one summary sentence Line 61-62.
Introduction: 1st paragraph still needs to identify a knowledge gap, what additional information does seeing the pattern across regions add to the debate/ knowledge about diel behavior	We edited the last sentence to focus on the knowledge gap. Lines: 71-72.
72: Use a different word than “illustrate” – makes you think of finger painting critters	We changed this sentence considering your next suggestion.
73. Run on sentence and very wordy here can be more concisely written as “Mammals occupy diverse diel niches due to many morphological, physiological and behavioral adaptations. These adaptations, including eye forms, sensorial systems, and endothermy evolved in response to biotic and abiotic factors”	Thanks for took in consideration your suggestion and changed this sentence Lines 73-74.
76-77 More concise way to communicate this would be to restructure to avoid passive voice “Endothermy facilitates nocturnal activity (colder time periods) and may have evolved in response to predation pressure form diurnal dinosaurs.”	We agreed and edited this sentence in the manuscript taking in consideration your suggestion Lines 77-79.
83-86 Run on sentence and very confusing – try to restructure for clarity	We modified this sentence to improve its understanding Lines 83-86.
92. remove “the temporal activity of” and “periods”	We removed it
94. remove “time of”	We removed it
106: What work is “Humid” doing in this sentence? Why not just “tropical forests”	We would prefer to keep the word “humid”, because there are dry tropical forests, where the environmental

	conditions can be different from humid tropical forest.
114-117 Run on. Can be broken in two sentences after (Fig 1).	Thanks for the suggestion. We broke it into two sentences Lines 114-117 .
124 Missing “such as”?	We included it
125 & 126 Change “that avoid” to “opposite” to allow for an objective observation. We don’t know that they are avoiding the species just because their behavior is opposite, we are inferring that.	Thanks, we changed it in the manuscript
126 Remove “furthermore”	We changed it in the manuscript
127 How would you infer that multiple alternative explanations are in play? Seems like this thought could be in the discussion instead of here.	Thanks, we removed it and used it partially in the discussion.
144 Change “the community” to “protected area”	We changed it in the manuscript
146 – 147 I know you added this analysis to appease my previous review, however it is not well integrated into the paper and does not address any of your questions as run.	We integrated these results into our manuscript and related them to our questions. Lines 152-156 .
Figure 2 I like this figure, however it sets up an expectation that you will see linear graphs/relationships in the results and how to interpret them. Instead you switch to different visual representations.	We changed the figure to be consistent with the displayed results.
Results: Hard to follow as they are presented – which test/model goes with which results. For example its unclear which model the results in lines 174-176 relate to and which hypothesis.	We incorporated the models employed to obtain the results.
180 Can clean this up as “We found a negative relationship in the neotropics with nocturnality decreasing with increasing body mass....”	Thanks, we changed it in the manuscript
Figure 3. Add which model these results came from – clogit or GLMM?	We added the type of model in the legend.
Figure 4. Add which model these results came from – clogit or GLMM?	We added the type of model in the legend.
233 Unclear what “This” refers to, avoid starting sentences with a “naked” adverb	We completed the sentence with “These results” Line 213 .
249-250 Very unclear what point this sentence is trying to convey, consider rewording	Here, our aim was to point out that there are many types of interactions, and some species may have higher predation risk than others, due to variable prey selection by predators. We acknowledge that the pattern we found may not be general for all species. There may be different degrees of avoidance.
286 – 287 How did you find that they overlapped “more than would predicted by chance” I see no evidence of a statistical test for this in the main text. Are you referring to the OverlapEst results?	We rephrased this sentence Lines 263-265 .
323 remove “activity”	Thanks, we removed it

376-377 Awkward sentence, rephrase	Yes, we rephrased it.
393 Missing "W" in "we"	Thanks, we corrected it
400 – 403 Fragment. Rephrase to move the rationale for removing elephants down in the paragraph (this does not need to be in the topic sentence).	We rephrased these it and move it down in the paragraph.